# Attitude Stabilization of a Satellite with Large Flexible Elements Using On-Board Actuators Only

Stepan Tkachev *[iD], Alexey Shestoperov, Anna Okhitina [iD] and Anna Nuralieva

Keldysh Institute of Applied Mathematics of the RAS, Miusskaya Sq. 4, 125047 Moscow, Russia;
alex.shestoperov@yandex.ru (A.S.); anna.ohitina@mail.ru (A.O.); annanuralieva@yandex.ru (A.N.)
* Correspondence: stevens_l@mail.ru

**Abstract:** Attitude control of a satellite with three flexible elements is considered. Control torque is developed by a set of reaction wheels, which are installed on the central hub of the satellite. The flexible elements are large, so the control torque constraints must be taken into account. In the paper, a control algorithm based on a linear-quadratic regulator is studied. The asymptotic stability of this control is shown. The choice of the control parameters is based on the closed form solution of the corresponding algebraic Riccati equation, which is supplemented by the linear matrix inequality. To increase the convergence rate, particle swarm optimization is used to tune the control parameters.

**Keywords:** attitude control; large flexible element; linear quadratic control; asymptotic stability; particle swarm optimization; control parameter

**MSC:** 93D20; 37N35; 37M05

## 1. Introduction

Consideration of elastic deformations is crucial to ensure high pointing precision for large space structures (for example, the ISS [1]), and, especially, for satellites with antennas [2], solar panels [3] and robotic manipulators [4]. Generally, the abovementioned elements have relatively low natural frequencies and damping ratios, so elastic deformations of such elements affect the whole system dynamics and could even lead to instabilities. Therefore, high-precision attitude control of flexible spacecraft has to suppress vibrations.

Numerous control design methods have been adapted and applied for attitude control of flexible spacecrafts. First of all, this concerns classical methods, such as proportional–integral–derivative (PID) controller [3], a linear quadratic regulator (LQR) [5] and sliding mode (SM) control [6]. In [7] attitude trajectory, the tracking problem is solved by means of LMI-based gain-scheduled H-infinity control. The SM control concept related to attitude maneuvers of a flexible spacecraft was further developed. In particular, Cao et al. [8] presented a robust attitude control law based on a novel nonsingular terminal sliding surface and operating in the face of actuator uncertainty and uncertain spacecraft dynamics. An SM adaptive fault-tolerant attitude tracking control algorithm is proposed for flexible spacecraft with partial loss of actuator effectiveness, unknown inertia parameters and external disturbances [9]. Additionally, using the singular perturbation theory [10,11], the equations of attitude motion of a flexible satellite are decomposed into fast and slow subsystems, and in both cases, the SM control laws are constructed relying on the latter one. Moreover, a novel component synthesis vibration suppression method is presented [12]. The authors note that the proposed active vibration suppression technique is consistent with such controllers as PD controllers, SM controllers and model predictive controls. The latter was used to accomplish large-angle single axis rotational maneuvers of a flexible satellite [13] and attitude/spin maneuvers of a spacecraft equipped with a large rotating flexible reflector (NASA Soil Moisture Passive Active mission) [2]. A number of research papers are devoted to the important issue of ensuring the high quality of transient processes during

attitude maneuvers [13,14]. In [14], a composite adaptive neural prescribed performance control ensuring the prescribed performance of transient response and attitude trajectory convergence in a preselected finite settling time is proposed. Fixed-time attitude control and stabilization using a neural network is considered in [15]. Fuzzy-logic optimal control was investigated in [16]. Thus, there are a huge variety of control approaches for flexible spacecraft attitude stabilization.

Control algorithms usually need parameter tuning. In the paper, particle swarm optimization (PSO) is used. In [17], the state-dependent Riccati equation (SDRE) attitude control technique is used within the scope of three flexible satellites formation flying, and the PSO algorithm tuned the pulse-width and pulse-frequency (PWPF) modulator, generating on–off thruster commands. The paper in [18] presented two combined LQR-PSO and SM-PSO attitude control methods, within which the PSO algorithm optimized the variable angle between the rigid hub and the payload. Note that not only the control law parameters appear as the subject of optimization. For instance, the kinematics of a rigid satellite undergoing constrained slew maneuvers using reaction wheels are tuned by an inverse-dynamics PSO approach [19].

Besides derivation of the attitude control law, there is the problem of obtaining its input information, such as attitude, angular velocity and the values of flexible deformations. The available set of actuators and sensors as well as their location obviously affects the choice of attitude control and vibration suppression strategies for a satellite with flexible appendages. In the last two decades, to solve the latter of these aforementioned problems, scientists have paid great attention to the use of smart structures, especially piezoelectric materials, as actuators or sensors [3,10,11,20–22]. For instance, Song and Agrawal [23] utilize PWPF modulation for producing the on–off thruster firing sequence required for attitude maneuver execution. At the same time, smart sensors and smart actuators with positive position feedback (PPF) control are used to actively suppress vibrations at the flexible appendage. However, the problem of simultaneous angular motion control of the satellite and vibration suppression in flexible elements, by means of actuators and sensors located only on the central body (hub), is still relevant. In this case, the clusters of momentum-exchanging devices are often employed as actuators, such as a system of four pyramid-mounted control moment gyros [1], ortho-skew construction of the four reaction wheels [24], and systems of six reaction wheels [12]. Concerning measuring devices and sensors for the motion estimation of spacecraft with flexible elements, Ivanov et al. [25] provided a relevant description as well as an overview of methods for the vibration determination. In [26], the comparison between Extended Kalman filter and Unscented Kalman filter is made through a Monte-Carlo simulation. In both cases, magnetometer and sun sensor measurements are used.

The attitude control in this paper is based on two assumptions that come from practical applications: the actuators and sensors are located on the central rigid body, and the data on the eigenmodes and eigen frequencies are rather inaccurate. Moreover, usually the performance of the on-board computer is relatively low, so the state vector cannot include a large number of variables. The principal idea is rather trivial—the control utilizes the rigid model of the spacecraft and ignores the flexibility. Thus, the control algorithm becomes as simple as possible for the on-board, almost real-time, implementation. It utilizes the estimation of the satellite position, attitude and angular velocity only. However, this simplicity creates a set of problems that be solved. First is the stability problem. The system model includes a large number of degrees of freedom, and the control effects directly only a few of them; moreover, most are completely ignored during the control calculation stage. So, a stability analysis of the system is needed, and in the paper, the theoretical basis of this approach is presented, and sufficient conditions are derived. Secondly, every algorithm has control parameters that allow one to tune the system behavior. Additionally, in the paper, the methodology for selection of the control parameters using PSO is provided.

In summary, this paper is devoted to the derivation of the attitude control law for a satellite with flexible appendages enabling low-frequency vibration suppression. After

the problem statement (Section 2), in Section 3 a mathematical model of a large flexible satellite, consisting of a satellite hub and three flexible elements—two solar arrays and an antenna—is presented. Flexible deformations of each element are described by the corresponding eigenmodes of vibration [27–29]. Section 4 contains linearized equations of satellite attitude motion. Sections 5 and 6 are dedicated to control synthesis methodology. It is assumed that actuators and sensors are located only on the hub, so the proposed control methodology does not require additional actuators on flexible elements. Also, it does not contain any information about modal amplitudes to avoid complicating the state estimation process and increasing the computational complexity of the control algorithm with their identification. Hence, a reduced (rigid) model of the spacecraft is used to obtain an attitude control law. In this context the following methods are tested: an inertia-free nonlinear attitude control algorithm derived by Sanyal et al. [30] and implemented by Posani et al. [24]; LQR [5] as well as SDRE techniques; the SDRE algorithm tuned by an input-shaping technique to reduce undesired elastic oscillations [31]. In the present paper, the LQR-PSO strategy is presented. The LQR is a well-known approach. However, in the present paper, the control is based on the reduced model (for a rigid body with fewer degrees of freedom) while being applied to the full one, which includes deformation modes. Direct implementation of the LQR in this case may lead to instability, so proper selection of the control parameters must be achieved. In Section 5, the sufficient conditions for asymptotic stability are derived, and an explicit solution of the algebraic Riccati equation (ARE) for this system is found. In Section 6, PSO searches for the optimal values of the LQR parameters minimizing the system's degree of stability. To improve the calculation effectiveness, the results of Section 5 are used. In Section 7, an illustration is presented. The appendices contain the d'Alembert principle for the whole system application and general forces calculation.

## 2. Problem Statement

The problem of attitude control of a satellite with three flexible elements (FE) is considered. The spacecraft is a rigid central hub with two flexible panels and an antenna cantilever attached to it (Figure 1).

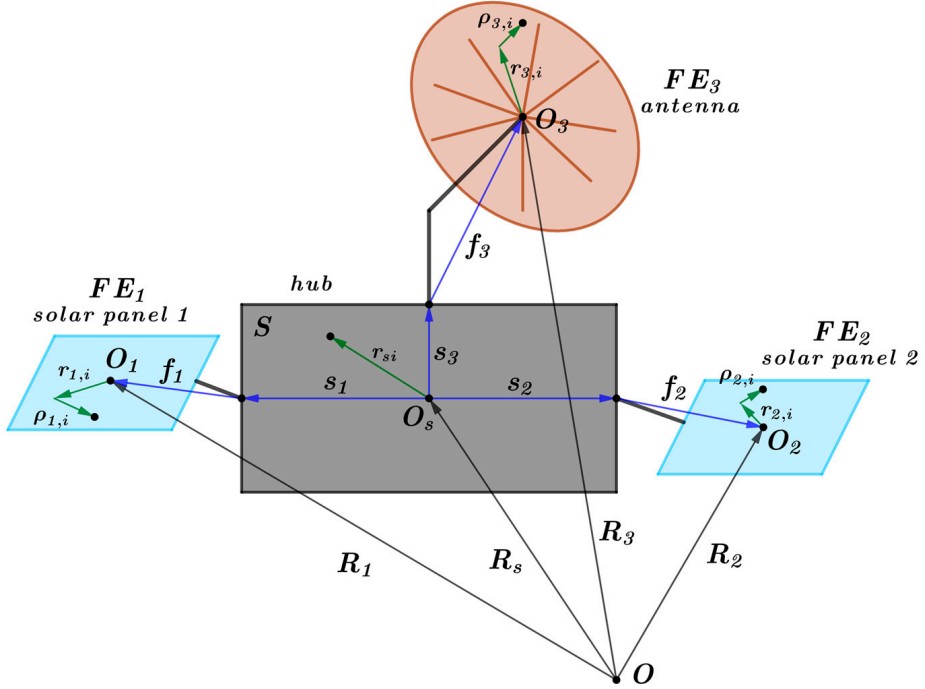

**Figure 1.** Satellite ($S$) with three flexible elements—solar panels ($FE_1$, $FE_2$) and antenna ($FE_3$).

The actuators (reaction wheels) and sensors (star tracker and angular velocity sensor) are installed on the central hub only. Therefore, there is neither direct action on nor direct measurements of the flexible deformations. The goal of the control system is to stabilize the hub in the inertial space and damp oscillations in the FEs.

## 3. Equations of Motion

There are two models, which are used in the paper: the full nonlinear model for the numerical modelling and linear model for the control algorithm synthesis, which is obtained from the first one. This section presents the nonlinear model.

The following reference frames are used:

1.  $OXYZ$ is the nonrotating frame; its origin coincides with Earth's center of mass, $OZ$ is perpendicular to the equatorial plane, $OX$ is directed to the vernal equinox point corresponding to a given epoch (e.g., J2000);
2.  $O_s xyz$ is the body-fixed frame; its origin lies in the satellite hub center of mass ($S$), and its axes coincide with its principal axes of inertia;
3.  $O_k x_k y_k z_k$, $k = \overline{1,3}$ are the flexible-element fixed frames with origin in the center of mass of the corresponding undeformed flexible element; axes are the principal axes of inertia of the undeformed flexible element.

The points of the satellite hub $m_{s,i}$, solar panel $m_{p,i}$ and antenna $m_{a,i}$ positions (in Figure 1) are defined as follows:

$$
\begin{aligned}
\mathbf{R}_{s,i} &= \mathbf{R}_s + \mathbf{r}_{s,i}, \\
\mathbf{R}_{k,i} &= \mathbf{R}_k + \mathbf{r}_{k,i} + \boldsymbol{\rho}_{k,i}, \ k = \overline{1,3},
\end{aligned}
\tag{1}
$$

where $\mathbf{R}_s$, $\mathbf{R}_k$ are the radius vectors of $S$ and $FE_k$ centers of mass, respectively, given in $OXYZ$, $\mathbf{r}_{s,i}$, $\mathbf{r}_{k,i}$ are the radius vectors of $S$ and $FE_k$ $i$-th points with respect to $O_s xyz$, $O_k x_k y_k z_k$, respectively, and $\boldsymbol{\rho}_{k,i}$ is the displacement of $FE_k$ $i$-th point due to deformation, respectively. It is assumed that $\boldsymbol{\rho}_{k,i} \ll \mathbf{r}_{k,i} \ll \mathbf{R}_k$, $k = \overline{1,3}$.

Normal modes are used for deformation definition. Point displacements due to deformations for each $FE_k$ are [27–29]

$$
\boldsymbol{\rho}_{k,i} = \mathbf{A}_{k,i} \mathbf{q}_k,
\tag{2}
$$

where $\mathbf{q}_k(t) = \left( q_1(t), \ \ldots \ , q_{n_k}(t) \right)^T$ are the vectors of modal coordinates, $n_k$ is the number of modes taken into account, $\mathbf{A}_{k,i}$ is the $3 \times n_k$ matrix of the mode shapes. The $j$-th column ($1 \le j \le n_k$) of these matrices defines the flexible displacements of the $i$-th point of the $FE_k$ (panel or antenna) caused by its $j$-th normal mode.

There are various approaches to derive the equations of motion of a flexible multibody system [27,32,33]. In this paper, the nonlinear model was developed using the d'Alembert [34] principle for each part of the satellite. The corresponding equations for the hull are [29,35]

$$
\mathbf{S}_s \begin{pmatrix} \ddot{\mathbf{R}}_s \\ \dot{\boldsymbol{\omega}}_s \end{pmatrix} = \mathbf{N}_s
\tag{3}
$$

where

$$
\mathbf{S}_s = \begin{pmatrix} m_s \mathbf{I}_{3\times 3} & 0 \\ 0 & \mathbf{J}_s \end{pmatrix}, \ \mathbf{N}_s = \begin{pmatrix} \mathbf{F}_s \\ \mathbf{M}_s - \boldsymbol{\omega}_s \times \mathbf{J}_s \boldsymbol{\omega}_s \end{pmatrix}.
\tag{4}
$$

Here, $\boldsymbol{\omega}_s$ is the absolute angular velocity of the hub, $\mathbf{F}_s$ and $\mathbf{M}_s$ are the net force and torques acting on the hub (including reaction forces in the joint points), respectively, and $m_s$ and $\mathbf{J}_s$ are the mass and the inertia tensor of the hub, respectively. $\mathbf{I}_{n\times n}$ is the n-by-n identity matrix.

The corresponding equations for the $FE_k$ are [29,35,36]

$$\mathbf{S}_k \begin{pmatrix} \ddot{\mathbf{R}}_k \\ \dot{\boldsymbol{\omega}}_k \\ \ddot{\mathbf{q}}_k \end{pmatrix} = \mathbf{N}_k, \tag{5}$$

where

$$\mathbf{S}_k = \begin{pmatrix} m_k \mathbf{I}_{3\times 3} & -[m_k \mathbf{A}_k \mathbf{q}_k]_\times & m_k \mathbf{A}_k \\ [m_k \mathbf{A}_k \mathbf{q}_k]_\times & \tilde{\mathbf{J}}_k & \sum\limits_{i\in FE_k} \left[ m_{k,i}\left(\mathbf{r}_{k,i}+\boldsymbol{\rho}_{k,i}\right)\right]_\times \mathbf{A}_{k,i} \\ m_k \mathbf{A}_k^T & -\sum\limits_{i\in FE_k} \mathbf{A}_{k,i}^T \left[ m_{k,i}\left(\mathbf{r}_{k,i}+\boldsymbol{\rho}_{k,i}\right)\right]_\times & \mathbf{I}_{n_k\times n_k} \end{pmatrix}, \tag{6}$$

$$\mathbf{N}_k = \begin{pmatrix} \mathbf{F}_k - m_k\boldsymbol{\omega}_k\times\boldsymbol{\omega}_k\times\mathbf{A}_k\mathbf{q}_k - 2m_k\boldsymbol{\omega}_k\times\mathbf{A}_k\dot{\mathbf{q}}_k \\ \sum\limits_{i\in FE_k}\left(\mathbf{r}_{k,i}+\boldsymbol{\rho}_{k,i}\right)\times\mathbf{F}_{k,i} - \boldsymbol{\omega}_k\times\tilde{\mathbf{J}}_k\boldsymbol{\omega}_k - 2\sum\limits_{i\in FE_k} m_{k,i}\left(\mathbf{r}_{k,i}+\boldsymbol{\rho}_{k,i}\right)\times\boldsymbol{\omega}_k\times\dot{\boldsymbol{\rho}}_{k,i} \\ -\boldsymbol{\Omega}_{n_k\times n_k}\mathbf{q}_k + \sum\limits_{i\in FE_k} m_{k,i}\mathbf{A}_{k,i}^T\mathbf{F}_{k,i} - \sum\limits_{i\in FE_k} m_{k,i}\mathbf{A}_{k,i}^T\boldsymbol{\omega}_k\times\boldsymbol{\omega}_k\times\left(\mathbf{r}_{k,i}+\boldsymbol{\rho}_{k,i}\right) - 2\sum\limits_{i\in FE_k} m_{k,i}\mathbf{A}_{k,i}^T\boldsymbol{\omega}_k\times\dot{\boldsymbol{\rho}}_{k,i} \end{pmatrix}. \tag{7}$$

Here, $\boldsymbol{\omega}_k$ is the absolute angular velocity of the $FE_k$, $\mathbf{F}_k$ and $\mathbf{M}_k$ are the net force and torques acting on the $FE_k$ (including reaction forces in the joint points), respectively, and $m_k$ and $\tilde{\mathbf{J}}_k$ are, respectively, the mass and the inertia tensor of the $FE_k$. The last one accounts for the deformations, also.

$$\mathbf{A}_k = \sum_{i\in FE_k} m_{k,i}\mathbf{A}_{k,i}. \tag{8}$$

$$\boldsymbol{\Omega}_{n_k\times n_k} = diag\left(\Omega_{k,1}^2 \quad \Omega_{k,2}^2 \quad \dots \quad \Omega_{k,n_k}^2\right) \tag{9}$$

$\Omega_{k,i}$ is the i-th eigen frequency of the $FE_k$. The notation $[\mathbf{r}]_\times$ for a vector $\mathbf{r} = \begin{pmatrix} r_1 & r_2 & r_3 \end{pmatrix}^T$ is

$$[\mathbf{r}]_\times = \begin{pmatrix} 0 & -r_3 & r_2 \\ r_3 & 0 & -r_1 \\ -r_2 & r_1 & 0 \end{pmatrix}.$$

Finally, the equation for the full state vector accelerations $\ddot{\mathbf{x}} = \begin{pmatrix} \ddot{\mathbf{R}}_s^T & \dot{\boldsymbol{\omega}}_s^T & \ddot{\mathbf{q}}_1^T & \ddot{\mathbf{q}}_2^T & \ddot{\mathbf{q}}_3^T \end{pmatrix}^T$ (the details can be found in the Appendix A) is

$$\tilde{\mathbf{S}}\ddot{\mathbf{x}} = \tilde{\mathbf{N}}, \tag{10}$$

where the total dynamics matrix $\tilde{\mathbf{S}}$ and right part vector $\tilde{\mathbf{N}}$ are

$$\tilde{\mathbf{S}} = \begin{pmatrix} \mathbf{S}_s + \sum\limits_{k=1}^{3} \mathbf{W}_{k,1}^T\mathbf{S}_k\mathbf{W}_{k,1} & \mathbf{W}_{11}^T\mathbf{S}_1\mathbf{W}_{12} & \mathbf{W}_{21}^T\mathbf{S}_2\mathbf{W}_{22} & \mathbf{W}_{31}^T\mathbf{S}_3\mathbf{W}_{32} \\ \mathbf{W}_{12}^T\mathbf{S}_1\mathbf{W}_{21} & \mathbf{W}_{12}^T\mathbf{S}_1\mathbf{W}_{12} & 0 & 0 \\ \mathbf{W}_{22}^T\mathbf{S}_2\mathbf{W}_{21} & 0 & \mathbf{W}_{22}^T\mathbf{S}_2\mathbf{W}_{22} & 0 \\ \mathbf{W}_{32}^T\mathbf{S}_3\mathbf{W}_{31} & 0 & 0 & \mathbf{W}_{32}^T\mathbf{S}_3\mathbf{W}_{32} \end{pmatrix}, \tag{11}$$

$$\tilde{\mathbf{N}} = \begin{pmatrix} \mathbf{N}_s + \sum\limits_{k=1}^{3}\left(\mathbf{W}_{k,1}^T\mathbf{N}_k - \mathbf{W}_{k,1}^T\mathbf{S}_k\mathbf{T}_k\right) \\ \mathbf{W}_{12}^T\mathbf{N}_1 - \mathbf{W}_{12}^T\mathbf{S}_1\mathbf{T}_1 \\ \mathbf{W}_{22}^T\mathbf{N}_2 - \mathbf{W}_{22}^T\mathbf{S}_2\mathbf{T}_2 \\ \mathbf{W}_{32}^T\mathbf{N}_3 - \mathbf{W}_{32}^T\mathbf{S}_3\mathbf{T}_3 \end{pmatrix}. \tag{12}$$

Here,

$$\mathbf{W}_{k,1} = \begin{pmatrix} \mathbf{E}_{3\times3} & -[\mathbf{s}_k + \mathbf{f}_k]_\times \\ \mathbf{0}_{3\times3} & \mathbf{I}_{3\times3} \\ \mathbf{0}_{n_k\times3} & \mathbf{0}_{n_k\times3} \end{pmatrix}, \quad \mathbf{W}_{k,2} = \begin{pmatrix} \mathbf{0}_{3\times n_k} \\ \mathbf{0}_{3\times n_k} \\ \mathbf{E}_{n_k\times n_k} \end{pmatrix}, \quad \mathbf{T}_k = \begin{pmatrix} \boldsymbol{\omega}_s \times \boldsymbol{\omega}_s \times (\mathbf{s}_k + \mathbf{f}_k) \\ \mathbf{0}_{1\times3} \\ \mathbf{0}_{1\times n_k} \end{pmatrix}. \quad (13)$$

Equation (10) is supplemented by the kinematic relations, which have the form

$$\begin{aligned} \dot{\mathbf{R}}_s &= \mathbf{V}_s, \\ \dot{\lambda}_{0s} &= -\tfrac{1}{2}(\boldsymbol{\omega}_s, \boldsymbol{\lambda}_s), \\ \dot{\boldsymbol{\lambda}}_s &= \tfrac{1}{2}(\lambda_{0s}\boldsymbol{\omega}_s + \boldsymbol{\lambda}_s \times \boldsymbol{\omega}_s), \\ \dot{\mathbf{q}}_k &= \mathbf{V}_{q_k}, \ k = \overline{1,3}, \end{aligned} \quad (14)$$

where $\left(\lambda_{0s} \quad \boldsymbol{\lambda}_s^T\right)^T$ is the attitude quaternion of the hub, $\mathbf{V}_{q_k}$ is the change rate vector of normal mode amplitudes for corresponding flexible elements $FE_k$. Equations (10) and (14) completely determine the motion of the considered system.

In comparison with [29], the current equations of motion are written relative to the center of mass of the rigid hub. In this case, if the number of flexible elements attached to the hub needs to be changed, the system (10) can be easily modified.

## 4. Linearized Mathematical Model

To derive the attitude control and study the asymptotic stability of the desired equilibrium, the linearized equations of satellite motion relative to its center of mass are used. Here, the required position

$$\begin{aligned} \boldsymbol{\omega}_s &= \mathbf{0}_{3\times1}, \ \boldsymbol{\lambda}_s = \mathbf{0}_{3\times1}, \\ \mathbf{q}_k &= \mathbf{0}_{n_k\times1}, \mathbf{V}_{\mathbf{q}_k} = \mathbf{0}_{n_k\times1}, \ k = \overline{1,3}, \end{aligned} \quad (15)$$

is considered. It is also supposed that there are no external torques and forces except control torque $\mathbf{u}$ produced by the set of actuators installed on the hub. The reasons for this assumption will be discussed in Section 5. Also, the orbital motion must be excluded. So, the system takes the following form:

$$\begin{pmatrix} \mathbf{S}_\omega & \mathbf{S}_{\omega q_1} & \mathbf{S}_{\omega q_2} & \mathbf{S}_{\omega q_3} \\ \mathbf{S}_{\omega q_1}^T & \mathbf{S}_{q_1} & \mathbf{S}_{q_1 q_2} & \mathbf{S}_{q_1 q_3} \\ \mathbf{S}_{\omega q_2}^T & \mathbf{S}_{q_1 q_2}^T & \mathbf{S}_{q_2} & \mathbf{S}_{q_2 q_3} \\ \mathbf{S}_{\omega q_3}^T & \mathbf{S}_{q_1 q_3}^T & \mathbf{S}_{q_2 q_3}^T & \mathbf{S}_{q_3} \end{pmatrix} \begin{pmatrix} \dot{\boldsymbol{\omega}}_s \\ \ddot{\mathbf{q}}_1 \\ \ddot{\mathbf{q}}_2 \\ \ddot{\mathbf{q}}_3 \end{pmatrix} = \begin{pmatrix} \mathbf{u} \\ -\boldsymbol{\Omega}_1 \mathbf{q}_1 \\ -\boldsymbol{\Omega}_2 \mathbf{q}_2 \\ -\boldsymbol{\Omega}_3 \mathbf{q}_3 \end{pmatrix}, \quad (16)$$

$$\begin{aligned} \mathbf{S}_\omega &= \mathbf{J} = \mathbf{J}_s + \sum_{k=1}^{3}\left(\mathbf{J}_k - m_k[\mathbf{s}_k + \mathbf{f}_k]_\times[\mathbf{s}_k + \mathbf{f}_k]_\times\right) + \tfrac{1}{m}\sum_{k=1}^{3}\left(m_k[\mathbf{s}_k + \mathbf{f}_k]_\times\right)\sum_{k=1}^{3}\left(m_k[\mathbf{s}_k + \mathbf{f}_k]_\times\right), \\ \mathbf{S}_{\omega q_k} &= \sum_{i\in FE_k} m_{k,i}[\mathbf{r}_{k,i}]_\times \mathbf{A}_{k,i} + m_k\left([\mathbf{s}_k + \mathbf{f}_k]_\times - \tfrac{1}{m}\sum_{k=1}^{3} m_k[\mathbf{s}_k + \mathbf{f}_k]_\times\right)\mathbf{A}_k, \ k = \overline{1,3}, \\ \mathbf{S}_{q_k} &= \mathbf{E}_{n_k\times n_k} - \tfrac{m_k^2}{m}\mathbf{A}_k^T\mathbf{A}_k, \ k = \overline{1,3}, \\ \mathbf{S}_{q_k q_l} &= -\tfrac{m_k m_l}{m}\mathbf{A}_k^T\mathbf{A}_l, \ k \neq l = \overline{1,3}. \end{aligned} \quad (17)$$

Grouping flexible variables as $\mathbf{q} = \left(\mathbf{q}_1^T \quad \mathbf{q}_2^T \quad \mathbf{q}_3^T\right)^T$, $\mathbf{V}_q = \left(\mathbf{V}_{q_1}^T \quad \mathbf{V}_{q_2}^T \quad \mathbf{V}_{q_3}^T\right)^T$ and $\boldsymbol{\Omega} = diag\left(\boldsymbol{\Omega}_1 \quad \boldsymbol{\Omega}_2 \quad \boldsymbol{\Omega}_3\right)$, and taking into account the kinematic Equation (14), gives

$$\begin{pmatrix} \mathbf{J} & \mathbf{S}_{\omega q} \\ \mathbf{S}_{\omega q}^T & \mathbf{S}_q \end{pmatrix} \begin{pmatrix} \dot{\boldsymbol{\omega}}_s \\ \dot{\mathbf{V}}_q \end{pmatrix} = -\begin{pmatrix} \mathbf{0} & \mathbf{0} \\ \mathbf{0} & \boldsymbol{\Omega} \end{pmatrix} \begin{pmatrix} \mathbf{0} \\ \mathbf{q} \end{pmatrix} + \begin{pmatrix} \mathbf{u} \\ \mathbf{0} \end{pmatrix}, \quad (18)$$

where

$$\mathbf{S}_{\omega q} = \begin{pmatrix} \mathbf{S}_{\omega q_1} & \mathbf{S}_{\omega q_2} & \mathbf{S}_{\omega q_3} \end{pmatrix}, \quad \mathbf{S}_q = \begin{pmatrix} \mathbf{S}_{q_1} & \mathbf{S}_{q_1 q_2} & \mathbf{S}_{q_1 q_3} \\ \mathbf{S}_{q_1 q_2}^T & \mathbf{S}_{q_2} & \mathbf{S}_{q_2 q_3} \\ \mathbf{S}_{q_1 q_3}^T & \mathbf{S}_{q_2 q_3}^T & \mathbf{S}_{q_3} \end{pmatrix}, \tag{19}$$

is the matrix of natural vibration frequencies of flexible elements.

Having solved the system (18) with respect to higher derivatives and using kinematics (14), the linear equations of angular motion are [37]

$$\dot{\mathbf{x}} = \mathbf{A}\mathbf{x} + \mathbf{B}\mathbf{u}, \tag{20}$$

where $\mathbf{x} = \begin{pmatrix} \boldsymbol{\omega}_s^T & \mathbf{V}_q^T & \boldsymbol{\lambda}_s^T & \mathbf{q}^T \end{pmatrix}^T$ is the state vector of SC with flexible elements,

$$\mathbf{A} = \begin{pmatrix} \mathbf{0}_{3\times3} & \mathbf{0}_{3\times n_\Sigma} & \mathbf{0}_{3\times3} & \mathbf{J}^{-1}\mathbf{S}_{\omega q}\left(\mathbf{S}_q - \mathbf{S}_{\omega q}^T\mathbf{J}^{-1}\mathbf{S}_{\omega q}\right)^{-1}\boldsymbol{\Omega} \\ \mathbf{0}_{n_\Sigma\times3} & \mathbf{0}_{n_\Sigma\times n_\Sigma} & \mathbf{0}_{n_\Sigma\times3} & -\left(\mathbf{S}_q - \mathbf{S}_{\omega q}^T\mathbf{J}^{-1}\mathbf{S}_{\omega q}\right)^{-1}\boldsymbol{\Omega} \\ \frac{1}{2}\mathbf{E}_{3\times3} & \mathbf{0}_{3\times n_\Sigma} & \mathbf{0}_{3\times3} & \mathbf{0}_{3\times n_\Sigma} \\ \mathbf{0}_{n_\Sigma\times3} & \mathbf{E}_{n_\Sigma\times n_\Sigma} & \mathbf{0}_{n_\Sigma\times3} & \mathbf{0}_{n_\Sigma\times n_\Sigma} \end{pmatrix},$$

$$\mathbf{B} = \begin{pmatrix} \mathbf{J}^{-1}\left(\mathbf{E}_{3x3} + \mathbf{S}_{\omega q}\left(\mathbf{S}_q - \mathbf{S}_q^T\mathbf{J}^{-1}\mathbf{S}_{\omega q}\right)^{-1}\mathbf{S}_{\omega q}^T\mathbf{J}^{-1}\right) \\ -\left(\mathbf{S}_q - \mathbf{S}_{\omega q}^T\mathbf{J}^{-1}\mathbf{S}_{\omega q}\right)^{-1}\mathbf{S}_{\omega q}^T\mathbf{J}^{-1} \\ \mathbf{0}_{3\times3} \\ \mathbf{0}_{n_\Sigma\times3} \end{pmatrix}. \tag{21}$$

Here, the number $n_\Sigma = \sum_{n=1}^3 n_k$ denotes the sum of all modes considered in the satellite's model. The latter system is used for the stabilization part of the control.

## 5. Control Synthesis

Since the control must stabilize in the inertial space, there will be two main external torques that affect the angular motion in the required equilibrium: gravity gradient and solar pressure. Both values are small (the first is about $10^{-3}\,\mathrm{N}\cdot\mathrm{m}$, the latter—$10^{-4}\,\mathrm{N}\cdot\mathrm{m}$) with respect to the possible control torque (maximum is $1\,\mathrm{N}\cdot\mathrm{m}$). The control consists of two parts: stabilizing and compensating. The second part compensates the external torques only, while the first one is based on the linear-quadratic regulator (LQR) [37] and does not include these torques. This control part is based on the reduced linearized model with no external torques and not requiring determination of the amplitudes of the eigenmodes to calculate the control of interest.

### 5.1. Stabilizing Control

The LQR is based on a linear solid-state model of the satellite, so in (20), the state vector and the matrices take the form

$$\mathbf{x} = \mathbf{x}_s = \begin{pmatrix} \boldsymbol{\omega}_s^T & \boldsymbol{\lambda}_s^T \end{pmatrix}^T, \quad \mathbf{A} = \mathbf{A}_s = \begin{pmatrix} \mathbf{0}_{3\times3} & \mathbf{0}_{3\times3} \\ \frac{1}{2}\mathbf{E}_{3\times3} & \mathbf{0}_{3\times3} \end{pmatrix}, \quad \mathbf{B} = \mathbf{B}_s = \begin{pmatrix} \mathbf{J}^{-1} \\ \mathbf{0}_{3\times3} \end{pmatrix}. \tag{22}$$

The LQR minimizes the following cost function [37]

$$I = \int_0^\infty \left(\mathbf{x}_s^T\mathbf{Q}\mathbf{x}_s + \mathbf{u}^T\mathbf{R}\mathbf{u}\right)dt \tag{23}$$

with positive definite matrices $\mathbf{Q}$, $\mathbf{R}$ that are the parameters of the algorithm. The LQR has the form [37]

$$\mathbf{u} = -\mathbf{R}^{-1}\mathbf{B}_s^T\mathbf{P}\mathbf{x}_s, \tag{24}$$

where $\mathbf{P}$ is a solution to the algebraic Riccati equation [37]

$$\mathbf{A}_s^T\mathbf{P} + \mathbf{P}\mathbf{A}_s - \mathbf{P}\mathbf{B}_s\mathbf{R}^{-1}\mathbf{B}_s^T\mathbf{P} + \mathbf{Q} = 0. \tag{25}$$

Since $\mathbf{A}_s$ is the $6 \times 6$ matrix, $\mathbf{P}$ size is also $6 \times 6$.

Since the pair $(\mathbf{A}, \mathbf{B})$ from (22) is controllable and $\mathbf{Q}$, $\mathbf{R}$ are positive definite matrices, the following are known [37]:

1. Matrix $\mathbf{P}$ from the LQR control law is the only positive definite solution of (25);
2. The LQR provides the asymptotic stability for the linear system with matrices (22).

Let the matrix $\mathbf{Q}$ have the form

$$\mathbf{Q} = \begin{pmatrix} \mathbf{Q}_{11} & \mathbf{Q}_{12} \\ \mathbf{Q}_{12}^T & \mathbf{Q}_{22} \end{pmatrix}, \tag{26}$$

where $\mathbf{Q}_{11}$ and $\mathbf{Q}_{22}$ are positive definite matrices ($\mathbf{Q}_{11} > 0$, $\mathbf{Q}_{22} > 0$). A positive definite solution of (25) $\mathbf{P}$ is convenient to represent as

$$\mathbf{P} = \begin{pmatrix} \mathbf{P}_{11} & \mathbf{P}_{12} \\ \mathbf{P}_{12}^T & \mathbf{P}_{22} \end{pmatrix}, \tag{27}$$

where $\mathbf{P}_{11} = \mathbf{P}_{11}^T$, $\mathbf{P}_{12}$, $\mathbf{P}_{22} = \mathbf{P}_{22}^T$ are $3 \times 3$ matrices. This representation is possible since $\mathbf{A}_s$ and $\mathbf{B}_s$ are block matrices with $3 \times 3$ elements. So, Equation (25) is rewritten as

$$\frac{1}{2}\left( \begin{pmatrix} \mathbf{P}_{12}^T & \mathbf{P}_{22} \\ \mathbf{0}_{3\times3} & \mathbf{0}_{3\times3} \end{pmatrix} + \begin{pmatrix} \mathbf{P}_{12} & \mathbf{0}_{3\times3} \\ \mathbf{P}_{22} & \mathbf{0}_{3\times3} \end{pmatrix} \right) - \begin{pmatrix} \mathbf{P}_{11}\mathbf{Z}\mathbf{P}_{11} & \mathbf{P}_{11}\mathbf{Z}\mathbf{P}_{12} \\ \mathbf{P}_{12}^T\mathbf{Z}\mathbf{P}_{11} & \mathbf{P}_{12}^T\mathbf{Z}\mathbf{P}_{12} \end{pmatrix} + \begin{pmatrix} \mathbf{Q}_{11} & \mathbf{Q}_{12} \\ \mathbf{Q}_{12}^T & \mathbf{Q}_{22} \end{pmatrix} = 0. \tag{28}$$

Here $\mathbf{Z} = \mathbf{J}^{-1}\mathbf{R}^{-1}\mathbf{J}^{-1}$ is the positive definite matrix since the inertia tensor is $\mathbf{J} > 0$.

As a result, Equation (25) is represented as a system of three equations:

$$\frac{1}{2}\left( \mathbf{P}_{12}^T + \mathbf{P}_{12} \right) - \mathbf{P}_{11}\mathbf{Z}\mathbf{P}_{11} + \mathbf{Q}_{11} = \mathbf{0}, \quad \frac{1}{2}\mathbf{P}_{22} = \mathbf{P}_{11}\mathbf{Z}\mathbf{P}_{12} + \mathbf{Q}_{12}, \quad \mathbf{P}_{12}^T\mathbf{Z}\mathbf{P}_{12} = \mathbf{Q}_{22}. \tag{29}$$

The fourth equation is exactly the same as the second equation in the system (29):

$$\mathbf{P}_{12}^T\mathbf{Z}\mathbf{P}_{11} + \mathbf{Q}_{12}^T = \frac{1}{2}\mathbf{P}_{22} = \frac{1}{2}\mathbf{P}_{22}^T = \left( \mathbf{P}_{12}^T\mathbf{Z}\mathbf{P}_{11} \right)^T = \mathbf{P}_{11}\mathbf{Z}\mathbf{P}_{12} + \mathbf{Q}_{12}. \tag{30}$$

LQR control with (22) taken into account is

$$\mathbf{u} = -\mathbf{R}^{-1}\mathbf{J}^{-1}(\mathbf{P}_{11}\boldsymbol{\omega}_s + \mathbf{P}_{12}\boldsymbol{\lambda}_s) = -\mathbf{K}_\omega\boldsymbol{\omega}_s - \mathbf{K}_\lambda\boldsymbol{\lambda}_s, \tag{31}$$

where $\mathbf{K}_\omega = \mathbf{R}^{-1}\mathbf{J}^{-1}\mathbf{P}_{11}$, $\mathbf{K}_\lambda = \mathbf{R}^{-1}\mathbf{J}^{-1}\mathbf{P}_{12}$. Since the control is based on the reduced system, it is necessary to select such matrices $\mathbf{K}_\omega$, $\mathbf{K}_\lambda$ that provide the asymptotic stability for the full system.

Consider the following Lyapunov function:

$$V = \frac{1}{2}\begin{pmatrix} \boldsymbol{\omega}_s^T & \mathbf{V}_q^T \end{pmatrix} \begin{pmatrix} \mathbf{J} & \mathbf{S}_{\omega q} \\ \mathbf{S}_{\omega q}^T & \mathbf{S}_q \end{pmatrix} \begin{pmatrix} \boldsymbol{\omega}_s \\ \mathbf{V}_q \end{pmatrix} + \boldsymbol{\lambda}_s^T\mathbf{K}_\lambda\boldsymbol{\lambda}_s + \frac{1}{2}\mathbf{q}^T\boldsymbol{\Omega}\mathbf{q}. \tag{32}$$

To satisfy $V > 0$, it is sufficient that $\mathbf{K}_\lambda > 0$. Its derivative due to the equations of motion is

$$\dot{V} = \boldsymbol{\omega}_s^T(\mathbf{u} + \mathbf{K}_\lambda\boldsymbol{\lambda}_s) = -\boldsymbol{\omega}_s^T\mathbf{K}_\omega\boldsymbol{\omega}_s. \tag{33}$$

If $\mathbf{K}_\omega > 0$, then $\dot{V} \leq 0$, and $\dot{V} = 0$ if and only if $\boldsymbol{\omega}_s = 0$. According to the Barbashin–Krasovski–LaSalle theorem [38], the equilibrium is asymptotically stable if there are no

other trajectories on the set $\left\{ \dot{V} = 0 \right\}$ besides the equilibrium. Such a set here is $\{ \boldsymbol{\omega}_s = 0 \}$; this implies

$$
\begin{aligned}
\mathbf{S}_{\omega q}\dot{\mathbf{V}}_q &= -\mathbf{K}_\lambda \boldsymbol{\lambda}_s, \\
\mathbf{S}_q\dot{\mathbf{V}}_q &= -\boldsymbol{\Omega}\mathbf{y}.
\end{aligned}
\tag{34}
$$

The right part of the first equation is a constant ($\boldsymbol{\lambda}_s = const$). In the second equation, the matrices $\mathbf{S}_q$ and $\boldsymbol{\Omega}$ are positive definite, so its solution can be represented as

$$
\mathbf{y} = \mathbf{H}\begin{pmatrix} C_1\cos(\nu_1 t + \varphi_{01}) \\ C_2\cos(\nu_2 t + \varphi_{02}) \\ \vdots \\ C_N\cos(\nu_N t + \varphi_{0N}) \end{pmatrix} = \mathbf{H}\boldsymbol{\Theta}(t),
\tag{35}
$$

where $\mathbf{H}$ consists of vectors $\mathbf{h}_i$ defined from the following equation:

$$
\left(\boldsymbol{\Omega} - \nu_i\mathbf{S}_q\right)\mathbf{h}_i = 0.
\tag{36}
$$

From the first Equation (34), it follows that

$$
\mathbf{S}_{\omega q}\mathbf{H}\ddot{\boldsymbol{\Theta}}(t) = -\mathbf{K}_\lambda\boldsymbol{\lambda}_s.
\tag{37}
$$

This equation is not satisfied for any values of the constants $C_i \neq 0$, $\varphi_{0i}$, if the frequencies $\nu_i$ are incommensurable and $rank(\mathbf{S}_{\omega q}\mathbf{H}) = 3$. Thus, the condition of the asymptotic stability of the zero solution is obtained. In practice, both conditions are usually fulfilled, but in some configurations (e.g., two identical panels installed symmetrically), this approach can face difficulties.

Thus, in the case of satisfying the abovementioned conditions for the satellite's dynamics, a sufficient condition for asymptotic stability is the positive definiteness of the quadratic forms $\mathbf{K}_\omega$, $\mathbf{K}_\lambda$. This means that $\mathbf{R}^{-1}$, $\mathbf{J}^{-1}$ and $\mathbf{P}_{11}$ should commute, since these matrices are positive definite. If $\mathbf{P}_{12}$ is positive definite, then $\mathbf{R}^{-1}$, $\mathbf{J}^{-1}$ and $\mathbf{P}_{12}$ should commute, too. For three positive definite quadratic forms to commute, it is necessary and sufficient that they have a diagonal form in the same basis. Since the inertia tensor $\mathbf{J}$ is defined, it determines the basis in which the remaining quadratic forms should have a diagonal form. In this case, $\mathbf{J} = \mathbf{W}\mathbf{J}^{diag}\mathbf{W}^T$, where $\mathbf{W}$ is an orthogonal matrix and $\mathbf{J}^{diag} = diag(J_1, J_2, J_3)$. Equation (20) with matrices (22) and, therefore, the system (29), has a unique solution $\mathbf{P} > 0$.

First, consider the case when $\mathbf{J}$ is diagonal. If $\mathbf{Q}$ and $\mathbf{R}$ are diagonal and their diagonal elements $R_i > 0$, $i = \overline{1,3}$, and $Q_i > 0$, $i = \overline{1,6}$, then the matrix $\mathbf{Z} = \mathbf{J}^{-1}\mathbf{R}^{-1}\mathbf{J}^{-1}$ is diagonal, i.e., $\mathbf{Z} = diag(z_1, z_2, z_3)$, where $z_i = 1/\left(R_i J_i^2\right)$, $i = \overline{1,3}$.

In this case, the solution of the system (29) consists of diagonal matrices $\mathbf{P}_{11}^{diag}$, $\mathbf{P}_{12}^{diag}$, $\mathbf{P}_{22}^{diag}$

$$
\mathbf{P}_{11}^{diag} = \begin{pmatrix} p'_1 & 0 & 0 \\ 0 & p'_2 & 0 \\ 0 & 0 & p'_3 \end{pmatrix},\ \mathbf{P}_{12}^{diag} = \begin{pmatrix} p_1 & 0 & 0 \\ 0 & p_2 & 0 \\ 0 & 0 & p_3 \end{pmatrix},\ \mathbf{P}_{22}^{diag} = \begin{pmatrix} p''_1 & 0 & 0 \\ 0 & p''_2 & 0 \\ 0 & 0 & p''_3 \end{pmatrix},
\tag{38}
$$

and it is positive definite, and hence the only stabilizing one. Indeed, a particular solution of the third equation from the system (29) is a matrix $\mathbf{P}_{12}$ with the following diagonal elements:

$$
p_i = \pm\sqrt{Q_{i+3}/z_i} = \pm J_i\sqrt{R_i Q_{i+3}},\ i = \overline{1,3}.
\tag{39}
$$

Then, from the first equation, the diagonal elements of matrix $\mathbf{P}_{11}$ are

$$
\begin{aligned}
p'_i &= \pm\sqrt{\frac{p_i + Q_i}{z_i}} = \pm\sqrt{\frac{\pm\sqrt{Q_{i+3}/z_i} + Q_i}{z_i}} = \\
&= \pm J_i\sqrt{R_i\left(\pm J_i\sqrt{R_i Q_{i+3}} + Q_i\right)},\ i = \overline{1,3},
\end{aligned}
\tag{40}
$$

and we obtain the diagonal elements of the matrix $\mathbf{P}_{22}$, substituting the coefficients $p_i$ and $p'_i$ into the second equation:

$$p''_i = 2z_i p'_i p_i = 2z_i \sqrt{\frac{Q_{i+3}}{z_i} \frac{p_i + Q_i}{z_i}} = 2\sqrt{Q_{i+3}(p_i + Q_i)} =$$
$$= 2\sqrt{q_{i+3}\left(\sqrt{Q_{i+3}/z_i} + Q_i\right)}, \ \ i = \overline{1,3}. \tag{41}$$

According to the Schur lemma [39], it is necessary that $\mathbf{P}_{11} > 0$ and $\mathbf{P}_{22} > 0$ in order for $\mathbf{P} > 0$. Hence, for the positive definiteness of the Riccati equation's solution, it is required to choose solutions with a plus sign when extracting the roots. It remains to check the fulfillment of the condition $\mathbf{P}_{22} - \mathbf{P}_{12}^T \mathbf{P}_{11}^{-1} \mathbf{P}_{12} > 0$ or $p''_i - (p_i)^2/p'_i > 0, \ i = \overline{1,3}$ in the case of diagonal matrices. Taking into account (39), (40) and (41), the latter becomes $\sqrt{Q_{i+3}/z_i} + 2Q_i > 0, \ i = \overline{1,3}$. This inequality is always satisfied because the coefficients $z_i, \ i = \overline{1,3}$ and $Q_i, \ i = \overline{1,6}$ are positive.

Thus, based on (39), (40) and (41), a positive definite solution of the algebraic Riccati equation is

$$\mathbf{P} = \begin{pmatrix} diag(p'_1, p'_2, p'_3) & diag(p_1, p_2, p_3) \\ diag(p_1, p_2, p_3) & diag(p''_1, p''_2, p''_3) \end{pmatrix}. \tag{42}$$

The linear quadratic control law (31) in this case has the form

$$\mathbf{u} = -diag\left[\left(\sqrt{\left(\sqrt{Q_{i+3}/R_i}J_i + Q_i/R_i\right)}\right)^3_{i=1}\right]\boldsymbol{\omega}_s - diag\left[\left(\sqrt{Q_{i+3}/R_i}\right)^3_{i=1}\right]\boldsymbol{\lambda}_s. \tag{43}$$

Now, consider the case when $\mathbf{J}$ is a non-diagonal positive definite matrix. Let the matrices $\mathbf{Q}_{11}$, $\mathbf{Q}_{22}$ and $\mathbf{R}$ be diagonal in the same basis as the inertia tensor, i.e.,

$$\begin{aligned} \mathbf{Q}_1 &= \mathbf{W}\mathbf{Q}_1^{diag}\mathbf{W}^T = \mathbf{W}diag(Q_1, Q_2, Q_3)\mathbf{W}^T, \\ \mathbf{Q}_2 &= \mathbf{W}\mathbf{Q}_2^{diag}\mathbf{W}^T = \mathbf{W}diag(Q_4, Q_5, Q_6)\mathbf{W}^T, \\ \mathbf{R} &= \mathbf{W}\mathbf{R}^{diag}\mathbf{W}^T = \mathbf{W}diag(R_1, R_2, R_3)\mathbf{W}^T, \end{aligned} \tag{44}$$

where the diagonal elements $r_i > 0, \ i = \overline{1,3}$ and $q_i > 0, \ i = \overline{1,6}$. The search for a stabilizing Riccati equation solution leads to the diagonal case. Particular solution of the third equation of the system (29) is sought in the form

$$\mathbf{P}_{12} = \mathbf{W}\mathbf{P}_{12}^{diag}\mathbf{W}^T \tag{45}$$

where $\mathbf{P}_{12}^{diag} = diag(p_1, p_2, p_3)$ is a diagonal matrix with the corresponding positive coefficients (39) of the diagonal inertia tensor case. Substitution of (45) into the first equation of the system (29) leads to the solution

$$\mathbf{P}_{11} = \mathbf{W}\mathbf{P}_{11}^{diag}\mathbf{W}^T, \tag{46}$$

where $\mathbf{P}_{11}^{diag} = diag(p'_1, p'_2, p'_3)$ is given by the expression (40). Similarly, the matrix $\mathbf{P}_{22}$ is calculated as

$$\mathbf{P}_{22} = \mathbf{W}\mathbf{P}_{22}^{diag}\mathbf{W}^T, \tag{47}$$

where the diagonal elements of the matrix $\mathbf{P}_{22}^{diag} = diag(p''_1, p''_2, p''_3)$ are given by the expression (41). Here, we use commutability of the diagonal matrices, the orthogonality of the matrix $\mathbf{W}$ and the fact that for an orthogonal matrix $\mathbf{W}$ and an arbitrary positive definite diagonal matrix $\Xi$, the matrix $\mathbf{W}\Xi\mathbf{W}^T$ is positive definite.

Then, the solution of ARE (29) is represented as

$$\mathbf{P} = \begin{pmatrix} \mathbf{W} & \mathbf{0}_{3\times3} \\ \mathbf{0}_{3\times3} & \mathbf{W} \end{pmatrix} \begin{pmatrix} \mathbf{P}_{11}^{diag} & \mathbf{P}_{12}^{diag} \\ \mathbf{P}_{12}^{diag} & \mathbf{P}_{22}^{diag} \end{pmatrix} \begin{pmatrix} \mathbf{W}^T & \mathbf{0}_{3\times3} \\ \mathbf{0}_{3\times3} & \mathbf{W}^T \end{pmatrix} \tag{48}$$

and it is positive definite. Herewith, the linear-quadratic control law (31)

$$\mathbf{u} = -\mathbf{R}^{-1}\mathbf{J}^{-1}\left(\mathbf{W}\mathbf{P}_{11}^{diag}\mathbf{W}^T\,\boldsymbol{\omega}_s + \mathbf{W}\mathbf{P}_{12}^{diag}\mathbf{W}^T\boldsymbol{\lambda}_s\right) = -\mathbf{K}_{\omega}\boldsymbol{\omega}_s - \mathbf{K}_{\lambda}\boldsymbol{\lambda}_s \tag{49}$$

stabilizes a linear system with matrices (22), and hence, the system with matrices (21). Moreover, since it is true for a linearized system, the same is true for the initial non-linear system also [38].

Thus, the possibility of stabilizing the system using the specified control law is shown, and the method for solving the Riccati equation for the considered problem is described.

*5.2. Compensation Control*

The provided stabilization control part ignores external torques. There are two torques that should be considered in the geostationary orbit for a satellite with large solar panels: solar radiation pressure and gravity gradient torque.

First, the general expression for external torques must be derived. The (12) has the form

$$\mathbf{N} = \begin{pmatrix} \mathbf{N}_s + \sum\limits_{k=1}^{3} \mathbf{W}_{k,1}^T \mathbf{N}_{k,} \\ \mathbf{W}_{12}^T \mathbf{N}_1 \\ \mathbf{W}_{22}^T \mathbf{N}_2 \\ \mathbf{W}_{32}^T \mathbf{N}_3 \end{pmatrix}, \quad \mathbf{N}_s = \begin{pmatrix} \mathbf{F}_s \\ \mathbf{M}_s \end{pmatrix}, \quad \mathbf{N}_k = \begin{pmatrix} \mathbf{F}_k \\ \sum\limits_{i\in FE_k} \left(\mathbf{r}_{k,i} + \boldsymbol{\rho}_{k,i}\right) \times \mathbf{F}_{k,i} \\ \sum\limits_{i\in FE_k} m_{k,i}\mathbf{A}_{k,i}^T\mathbf{F}_{k,i} \end{pmatrix}, \quad k = \overline{1,3}. \tag{50}$$

Since

$$\mathbf{W}_{k,1}^T \mathbf{N}_k = \begin{pmatrix} \mathbf{F}_k \\ [\mathbf{s}_k + \mathbf{f}_k]_\times \mathbf{F}_k + \sum\limits_{i\in FE_k}\left(\mathbf{r}_{k,i} + \boldsymbol{\rho}_{k,i}\right) \times \mathbf{F}_{k,i} \end{pmatrix}, \quad k = \overline{1,3}, \tag{51}$$

$$\mathbf{W}_{k,2}^T \mathbf{N}_k = \sum\limits_{i\in FE_k} m_{k,i}\mathbf{A}_{k,i}^T\mathbf{F}_{k,i}, \quad k = \overline{1,3} \tag{52}$$

The resulting general force vector is

$$\mathbf{N} = \begin{pmatrix} \mathbf{M}_s + \sum\limits_{k=1}^{3}\left([\mathbf{s}_k + \mathbf{f}_k]_\times \mathbf{F}_k + \sum\limits_{i\in FE_k}\left(\mathbf{r}_{k,i} + \boldsymbol{\rho}_{k,i}\right) \times \mathbf{F}_{k,i}\right) - \frac{1}{m}\left(\sum\limits_{k=1}^{3} m_k[\mathbf{s}_k + \mathbf{f}_k]_\times\right)\mathbf{F} \\ \sum\limits_{i\in FE_1} m_{1,i}\mathbf{A}_{1,i}^T\mathbf{F}_{1,i} - \frac{1}{m}m_1\mathbf{A}_1^T\mathbf{F} \\ \sum\limits_{i\in FE_2} m_{2,i}\mathbf{A}_{2,i}^T\mathbf{F}_{2,i} - \frac{1}{m}m_2\mathbf{A}_2^T\mathbf{F} \\ \sum\limits_{i\in FE_3} m_{3,i}\mathbf{A}_{3,i}^T\mathbf{F}_{3,i} - \frac{1}{m}m_3\mathbf{A}_3^T\mathbf{F} \end{pmatrix}, \tag{53}$$

where

$$\mathbf{F} = \mathbf{F}_s + \sum\limits_{k=1}^{3} \mathbf{F}_k \tag{54}$$

is the net force acting upon the system. The first component of (53) is the net torque of all forces acting upon the whole system. The torques, which are taken into account in the compensation part of the control, are $\mathbf{M}^{grav}$ and $\mathbf{M}^{sun}$. The first one corresponds to the rigid body gravity gradient torque (see Appendix B) [40]

$$\mathbf{M}^{grav} = 3\frac{\mu}{R^3}\mathbf{R} \times \mathbf{J}\mathbf{R}. \tag{55}$$

In this case, the resulting control torque is

$$\mathbf{M}^{ctrl} = -\mathbf{M}^{grav} - \mathbf{M}^{sun} - \mathbf{u}. \tag{56}$$

The solar radiation torque for symmetrical configuration of the panels is thought to be near zero and is considered as a perturbation due to the small difference between the panel and its mounting point. This control law is used for the spacecraft stabilization in the neighborhood of the required position.

## 6. Optimization Problem

LQR requires the choice of control parameters, i.e., diagonalized matrices $\mathbf{Q}_{11}$, $\mathbf{Q}_{22}$ and $\mathbf{R}$. These matrices significantly affect the maximum values of the control vector components and the quality of the algorithm. Here the degree of stability [41] is taken as a quality metric. The maximum affordable control torque value is the upper bound. For linear systems with constant coefficients whose equilibrium position is asymptotically stable, the degree of stability is the distance from the imaginary axis to the rightmost root of the characteristic equation. In fact, this is the exponent with the least damping.

A closed-loop system with control

$$\mathbf{u} = -\mathbf{Kx}, \tag{57}$$

where $\mathbf{K} = \begin{pmatrix} \mathbf{K}_\omega & \mathbf{0}_{3 \times n} & \mathbf{K}_\lambda & \mathbf{0}_{3 \times n} \end{pmatrix}$, is represented in the form

$$\dot{\mathbf{x}} = (\mathbf{A} - \mathbf{BK})\mathbf{x} = \mathbf{A}_c \mathbf{x}. \tag{58}$$

Eigenvalues of matrix $\mathbf{A}_c$ determine the transient process rate. In this case, it is also necessary to take into account the constraint on control

$$\max |\mathbf{Kx}| \leq \mathbf{u}_{\max}. \tag{59}$$

Thus, the optimization problem is

$$\mathbf{\Phi} = -\mathrm{Re}\mu_{\min} \rightarrow \max \text{ under } \max |\mathbf{Kx}| \leq \mathbf{u}_{\max}. \tag{60}$$

Here, $\mu_{\min}$ is the eigenvalue with the minimum distance to the imaginary axis on the complex plane of characteristic values.

It is necessary to formalize the criterion (59), since as a rule, there is a surge effect, i.e., an increase in the required control at the beginning of transients. For this, the approach described in [42] is used. The Lyapunov function (32) is

$$V = \frac{1}{2} \begin{pmatrix} \boldsymbol{\omega}_s^T & \mathbf{V}_q^T & \boldsymbol{\lambda}_s^T & \mathbf{q}^T \end{pmatrix} \begin{pmatrix} \mathbf{S}_\omega & \mathbf{S}_{\omega q} & \mathbf{0}_{3 \times 3} & \mathbf{0}_{3 \times N} \\ \mathbf{S}_{\omega q}^T & \mathbf{S}_q & \mathbf{0}_{N \times 3} & \mathbf{0}_{N \times N} \\ \mathbf{0}_{3 \times 3} & \mathbf{0}_{3 \times N} & 2\mathbf{K}_\lambda & \mathbf{0}_{N \times N} \\ \mathbf{0}_{N \times 3} & \mathbf{0}_{N \times N} & \mathbf{0}_{N \times N} & \boldsymbol{\Omega} \end{pmatrix} \begin{pmatrix} \boldsymbol{\omega}_s \\ \mathbf{V}_q \\ \boldsymbol{\lambda}_s \\ \mathbf{q} \end{pmatrix} = \frac{1}{2} \mathbf{x}^T \mathbf{Hx}. \tag{61}$$

At the initial moment $t = t_0$, we have $0.5 \, \mathbf{x}_0^T \mathbf{Hx}_0 = a_0$, where $\mathbf{x}_0 = \mathbf{x}(t_0)$. Due to the decreasing Lyapunov function, we obtain $0.5 \, \mathbf{x}^T \mathbf{Hx} \leq a_0$ or

$$\frac{1}{2a_0} \mathbf{x}^T \mathbf{Hx} \leq 1. \tag{62}$$

Rewrite the condition (59) in the form

$$\mathbf{x}^T \mathbf{K}^T \mathbf{Kx} \leq \mathbf{u}_{\max}^2. \tag{63}$$

Thus, it is necessary to guarantee (63) under the condition (62). This is true if and only if the matrix inequality

$$\frac{1}{u_{\max}^2}\mathbf{K}^T\mathbf{K} \leq \frac{1}{2a_0}\mathbf{H} \tag{64}$$

is satisfied. By Schur's lemma, this is equivalent to the fact that the following matrix is positive definite.

$$\begin{pmatrix} \mathbf{H} & \mathbf{K}^T \\ \mathbf{K} & \frac{\mathbf{u}_{\max}^2}{2a_0}\mathbf{E}_{3\times 3} \end{pmatrix} \geq 0. \tag{65}$$

So, if the linear feedback matrix $\mathbf{K}$ satisfies (65), the constraint will not be violated, and simultaneously, due to the decreasing (61), the system will not leave the initial ball (62).

In this case, the cost function and restrictions have complex form and cannot be presented explicitly, depending on the problem parameters. To solve this problem, the evolutionary optimization method—particle swarm optimization (PSO) [43]—is implemented for the search of the set of optimal control parameters.

Let $\mathbf{x}_p$ be a set of the control parameters. The PSO is based on the decision-making model of each swarm particle. The model describing the decision making of particles in a swarm turned out to be a simple and effective optimization method. The task of the swarm is to provide a minimum of the given cost function

$$\boldsymbol{\Phi}\big(\mathbf{x}_p\big) : \mathbb{R}^D \to \mathbb{R} \tag{66}$$

on the search domain

$$\mathbb{U} = \left\{ \mathbf{x}_p \in \mathbb{R}^D \ \middle| \ \eta_{low,j} \leq x_{p,j} \leq \eta_{up,j}, \ j = \overline{1,D} \right\} \tag{67}$$

defined by restrictions on the values of the $D$ parameters. Each particle $p = \overline{1,P}$ in each generation $i = \overline{1,G}$ has a certain position $\mathbf{x}_p(i)$ and velocity $\mathbf{v}_p(i)$. The position of the particle specifies a possible solution to the optimization problem.

The velocity allows deciding the direction of displacement to continue the search and consists of three components:

$$\mathbf{v}_p(i) = c_{in}\mathbf{v}_p(i-1) + c_{cog}\Big(\mathbf{x}_{p,best}(i) - \mathbf{x}_p(i)\Big) + c_{soc}\Big(\mathbf{x}_{p,local\,best}(i) - \mathbf{x}_p(i)\Big). \tag{68}$$

The position of each particle at the next iteration is determined based on its current position and velocity:

$$\mathbf{x}_p(i+1) = \mathbf{x}_p(i) + \mathbf{v}_p(i) \tag{69}$$

The first term in (68) is the inertial component; it is responsible for the search continuation in the same direction. The second is the cognitive component; the particle desires to return to its own best position found earlier, $\mathbf{x}_{p,best}$. The last one is the social component, which represents striving for the best position found in the particle vicinity, $\mathbf{x}_{p,local\,best}$.

The value of the cost function $\boldsymbol{\Phi}_{p,local\,best}$ corresponds to the best particle $p$ position $\mathbf{x}_{p,best}$, $\boldsymbol{\Phi}_{p,local\,best}$ corresponds to the best position in the vicinity of the particle $p$ $\mathbf{x}_{p,local\,best}$:

$$\begin{aligned} \boldsymbol{\Phi}_{p,best} &= \boldsymbol{\Phi}\Big(\mathbf{x}_{p,best}\Big) \\ \boldsymbol{\Phi}_{p,local\,best} &= \boldsymbol{\Phi}\Big(\mathbf{x}_{p,local\,best}\Big)' \end{aligned} \tag{70}$$

and $\boldsymbol{\Phi}_{best}$ is the global optimal solution found by the entire swarm in $i$ iterations.

The contribution of each velocity component is varied using appropriate weighting coefficients $c_{in}$, $c_{cog}$, $c_{soc}$. A large value of the inertial coefficient $c_{in}$ accelerates the exploration of search space and does not allow it to fall into local optima. The correct selection of social $c_{soc}$ and cognitive $c_{cog}$ coefficients allows each particle to first look for its best position, and then switch attention to improving the best position found among all of the

particle's neighbors. For each optimization problem, the coefficients should be selected individually, focusing on some empirical rules and selection methods, which are given, for example, in [43]. However, in any problem when selecting coefficients, it is necessary that the following relationship be satisfied

$$c_{in} > \frac{1}{2}\left(c_{soc} + c_{cog}\right) - 1 \tag{71}$$

to ensure the convergence of the PSO, which was shown in [44].

The search stop criterion is the fulfillment of the following conditions simultaneously:

1. the cost function derivative is small (dimensionless parameter of cost function stagnation is $\Phi_{stagn}$);
2. all particles are falling into some neighborhood of the best position (dimensionless parameter of swarm stagnation is $S_{stagn}$).

Figure 2 shows a block diagram of the described algorithm for clarity.

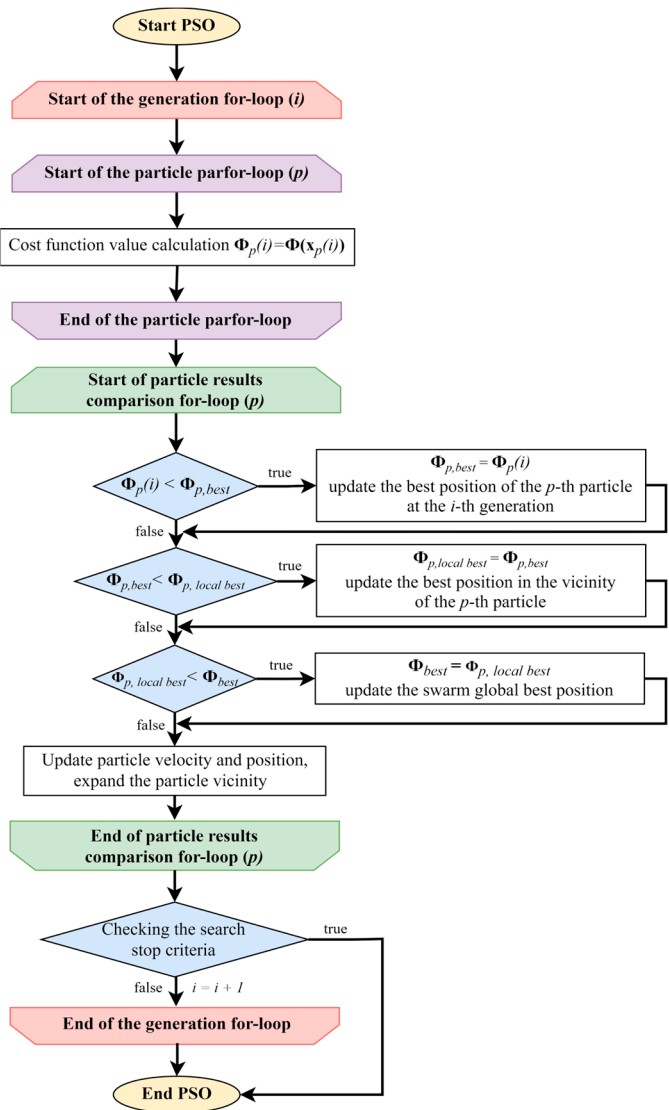

**Figure 2.** Block diagram of the PSO algorithm.

As a result, PSO searches for the optimal values of the LQR parameters, taking into account the fulfillment of condition (65). Minimization of the degree of stability can be carried out with a different number of known modes.

Since matrices $\mathbf{Q}_{11}$, $\mathbf{Q}_{22}$ and $\mathbf{R}$ are diagonal, there are nine parameters that should be found. However, matrices $\mathbf{K}_\omega$ and $\mathbf{K}_\lambda$ depend on the ratios of $\mathbf{Q}_{11}$ and $\mathbf{Q}_{22}$ to $\mathbf{R}$, so one of these matrices can be fixed. Consider that $\mathbf{R} = diag\begin{pmatrix} 1 & 1 & 1 \end{pmatrix}$ in the basis of principle axes of the system inertia tensor. The other six parameters are defined by the PSO method. So $\mathbf{x}_p$ (66) is $\mathbf{x}_p = \begin{pmatrix} Q_1 & Q_2 & Q_3 & Q_4 & Q_5 & Q_6 \end{pmatrix}_p$, which are presented in (44).

Since (65) depends on the initial condition, the LQR parameters will also depend on the initial state. So, two sets of PSO bounds are used to improve the convergence in the neighborhood of the equilibrium. The first domain is

$$\begin{aligned} |\omega_i| &\leq 10^{-3} rad/sec, \\ |\lambda_i| &\leq 0.5; \end{aligned} \tag{72}$$

the second one is

$$\begin{aligned} |\omega_i| &\leq 10^{-6} rad/sec, \\ |\lambda_i| &\leq 3 \times 10^{-4}. \end{aligned} \tag{73}$$

It is considered that satellites are deployed with rather small angular momentum, so the initial domain for the angular velocity is about 10 times more than the orbital angular velocity of the geostationary orbit, which is $\omega_{orb} = 0.7 \times 10^{-5} \, rad/sec$. Such selection of the initial condition domains is based on the following practical consideration. The spacecraft with large FE usually separates from the launch vehicle with rather small initial angular velocity, and its FE are undeployed. The deployment of FE increases the inertia tensor by two orders and hence decreases the angular velocity by the same value due to the conservation of angular momentum. The final domain is about 10 times less than the $\omega_{orb}$. The control limit is $u_{\max} = 1 \, N \cdot m$. PSO parameters are given in Table 1.

**Table 1.** Main parameters of the particle swarm method.

| $D$ | $D = 6$ | |
|---|---|---|
| $\eta_{low}$ and $\eta_{up}$ | $\eta_{low}^1 = [10^2 \cdot (1,1,1), \ 0.1 \cdot (1,1,1)]$ $\eta_{up}^1 = [10^5 \cdot (1,1,1), \ 10^2 \cdot (1,1,1)]$ | $\eta_{low}^2 = [10^6 \cdot (1,1,1), \ 10^5 \cdot (1,1,1)]$ $\eta_{up}^2 = [10^9 \cdot (1,1,1), \ 10^8 \cdot (1,1,1)]$ |
| $P$ | 200 | |
| $G$ | 500 | |
| $c_{in}^{up}, \ c_{in}^{low}$ | 0.9, 0.4 | |
| $c_{cog}^{up}, \ c_{cog}^{low}$ | 2.05, 0 | |
| $c_{soc}^{up}, \ c_{soc}^{low}$ | 2.05, 0 | |
| $\Phi_{stagn}$ | 0.001 | |
| $S_{stagn}$ | 0.005 | |

So, the two following sets of LQR parameters are found:

$$\mathbf{Q}_{11}^1 = \begin{pmatrix} 8.49 \times 10^5 & -0.02 & 0.85 \\ -0.02 & 8.18 \times 10^5 & 4.32 \times 10^3 \\ 0.85 & 4.32 \times 10^3 & 4.40 \times 10^3 \end{pmatrix} (\mathrm{N} \cdot \mathrm{m} \cdot \mathrm{s})^2, \ \mathbf{Q}_{22}^1 = \begin{pmatrix} 0.45 & 0 & 0 \\ 0 & 0.43 & 0.02 \\ 0 & 0.02 & 0.1 \end{pmatrix} (\mathrm{N} \cdot \mathrm{m})^2 \tag{74}$$

$$\mathbf{Q}_{11}^2 = \begin{pmatrix} 4.34 \times 10^7 & 17.9 & 444 \\ 17.9 & 3.92 \times 10^7 & 2.12 \times 10^6 \\ 444 & 2.12 \times 10^6 & 1.18 \times 10^6 \end{pmatrix} (\mathrm{N} \cdot \mathrm{m} \cdot \mathrm{s})^2, \ \mathbf{Q}_{22}^2 = \begin{pmatrix} 2.31 \times 10^5 & 1.35 & 0.54 \\ 1.35 & 1.00 \times 10^5 & -4.79 \times 10^3 \\ 0.54 & -4.79 \times 10^3 & 1.86 \times 10^5 \end{pmatrix} (\mathrm{N} \cdot \mathrm{m})^2 \tag{75}$$

The corresponding feedback matrices $\mathbf{K}_\omega$ and $\mathbf{K}_\lambda$ are

$$\mathbf{K}_\omega^1 = \begin{pmatrix} 412 & 0 & 0 \\ 0 & 404 & 16 \\ 0 & 16 & 120 \end{pmatrix} \mathrm{N} \cdot \mathrm{m} \cdot \mathrm{s}, \ \mathbf{K}_\lambda^1 = \begin{pmatrix} 0.67 & 0 & 0 \\ 0 & 0.66 & 0 \\ 0 & 0 & 0.32 \end{pmatrix} \mathrm{N} \cdot \mathrm{m} \tag{76}$$

$$\mathbf{K}_\omega^2 = \begin{pmatrix} 1.02 & 0 & 0 \\ 0 & 0.89 & 0.03 \\ 0 & 0.03 & 0.39 \end{pmatrix} \times 10^4 \, \text{N} \cdot \text{m} \cdot \text{s}, \quad \mathbf{K}_\lambda^2 = \begin{pmatrix} 480 & 0 & 0 \\ 0 & 317 & -6 \\ 0 & -6 & 431 \end{pmatrix} \text{N} \cdot \text{m} \quad (77)$$

Both the values of matrices $\mathbf{Q}_{11}$, $\mathbf{Q}_{22}$ and $\mathbf{K}_\omega$, $\mathbf{K}_\lambda$ are rather large, which may lead to large control efforts for some domains of the initial conditions. However, the sets of these parameters are found in such a way that for any initial conditions in the domains (72) and (73), the control constraint (59) can not be violated. So, the control will always be less than feasible $u_{\max} = 1 \, \text{N} \cdot \text{m}$.

These two sets are used in the following section, where the algorithm operation is demonstrated.

The crucial proof for the stability is the fact that the control matrices and inertia tensor are diagonal in the same basis. The problem here is the fact that the inertia tensor used in the model can (and likely) differs from the one in the real system. Let $\mathbf{J}_0$ be the nominal inertia tensor that is used in the mathematical model and $\mathbf{J} = \mathbf{J}_0(\mathbf{E}_{3\times3} + \varepsilon\mathbf{j})$ be the real one, where $\varepsilon$ is a small parameter that shows the difference between the nominal and real tensor and $\mathbf{j}$ is the symmetric matrix, the norm of which in some sense is one.

Consider matrix $\mathbf{K}_\omega$ ($\mathbf{K}_\lambda$ is analogous)

$$\begin{aligned} \mathbf{K}_\omega = \mathbf{R}^{-1}\mathbf{J}^{-1}\mathbf{P}_{11} = \mathbf{R}^{-1}(\mathbf{E}_{3\times3} + \varepsilon\mathbf{j})^{-1}\mathbf{J}_0^{-1}\mathbf{P}_{11} \approx \\ \approx \mathbf{R}^{-1}(\mathbf{E}_{3\times3} - \varepsilon\mathbf{j})\mathbf{J}_0^{-1}\mathbf{P}_{11} = \mathbf{R}^{-1}\mathbf{J}_0^{-1}\mathbf{P}_{11} - \varepsilon\mathbf{R}^{-1}\mathbf{j}\mathbf{J}_0^{-1}\mathbf{P}_{11}. \end{aligned} \quad (78)$$

The first term is the positive definite matrix, while the second one in the general case can be even nonsymmetrical. However, the equivalent symmetric form matrix is

$$\boldsymbol{\omega}^T\mathbf{R}^{-1}\mathbf{j}\mathbf{J}_0^{-1}\mathbf{P}_{11}\boldsymbol{\omega} = \frac{1}{2}\boldsymbol{\omega}^T\left(\mathbf{R}^{-1}\mathbf{j}\mathbf{J}_0^{-1}\mathbf{P}_{11} + \mathbf{P}_{11}^T\mathbf{J}_0^{-1}\mathbf{j}\mathbf{R}^{-1}\right)\boldsymbol{\omega}, \quad (79)$$

so

$$\mathbf{K}_\omega \approx \mathbf{R}^{-1}\mathbf{J}_0^{-1}\mathbf{P}_{11} - \frac{1}{2}\varepsilon\left(\mathbf{R}^{-1}\mathbf{j}\mathbf{J}_0^{-1}\mathbf{P}_{11} + \mathbf{P}_{11}^T\mathbf{J}_0^{-1}\mathbf{j}\mathbf{R}^{-1}\right). \quad (80)$$

The components of this matrix in the basis of principal axes of $\mathbf{J}_0$ are

$$\mathbf{K}_\omega = \begin{pmatrix} k_{11}^0 - \varepsilon k_{11}^1 & -\varepsilon k_{12}^1 & -\varepsilon k_{13}^1 \\ -\varepsilon k_{12}^1 & k_{22}^0 - \varepsilon k_{22}^1 & -\varepsilon k_{23}^1 \\ -\varepsilon k_{12}^1 & -\varepsilon k_{23}^1 & k_{33}^0 - \varepsilon k_{33}^1 \end{pmatrix}. \quad (81)$$

Since $k_{ii}^0 > 0$, $k_{ij}^0$ and $k_{ij}^1$ have the same order while $\varepsilon$ is a small parameter (the error is usually small), then the matrix $\mathbf{K}_\omega$ is positive definite for the sufficiently small $\varepsilon$.

## 7. Numerical Example

To demonstrate the typical system behavior, a numerical example is presented (Figures 3–8). The system parameters for numerical simulation are presented in Table 2.

The numerical modelling is performed in the nonlinear model (10)–(14). Two sets of control parameters are taken. The simulation results are shown in the following figures.

Figures show that the control stabilizes the satellite and decreases the modal variable amplitudes. The process is rather slow since the control is small with respect to the total inertia tensor of the satellite with flexible elements. The peaks in Figures 5 and 6 correspond to the switching between the sets of control parameters. As one can see from Figures 7 and 8, this allows an increase in the convergence rate. The control level is almost three times less than the threshold. This is due to the fact that the approach guarantees (63) for each initial condition set in the domain (72). The evolution of the reaction wheel angular momentum in Figure 6 is due to the gravity gradient torque compensation.

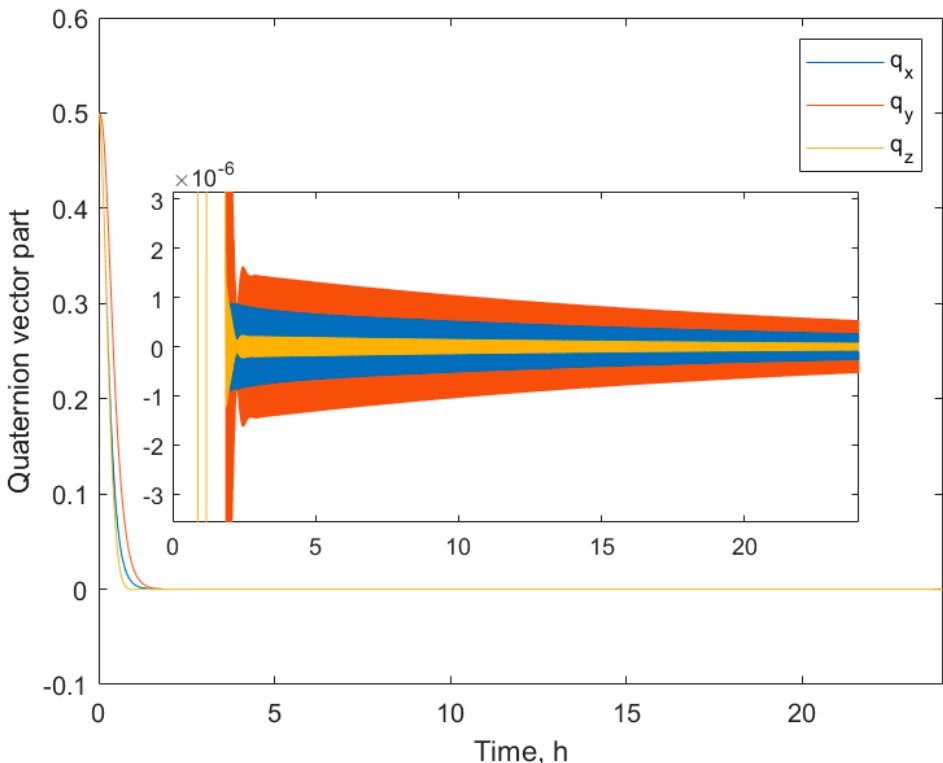

**Figure 3.** Vector part of the attitude quaternion of the hub (*S*).

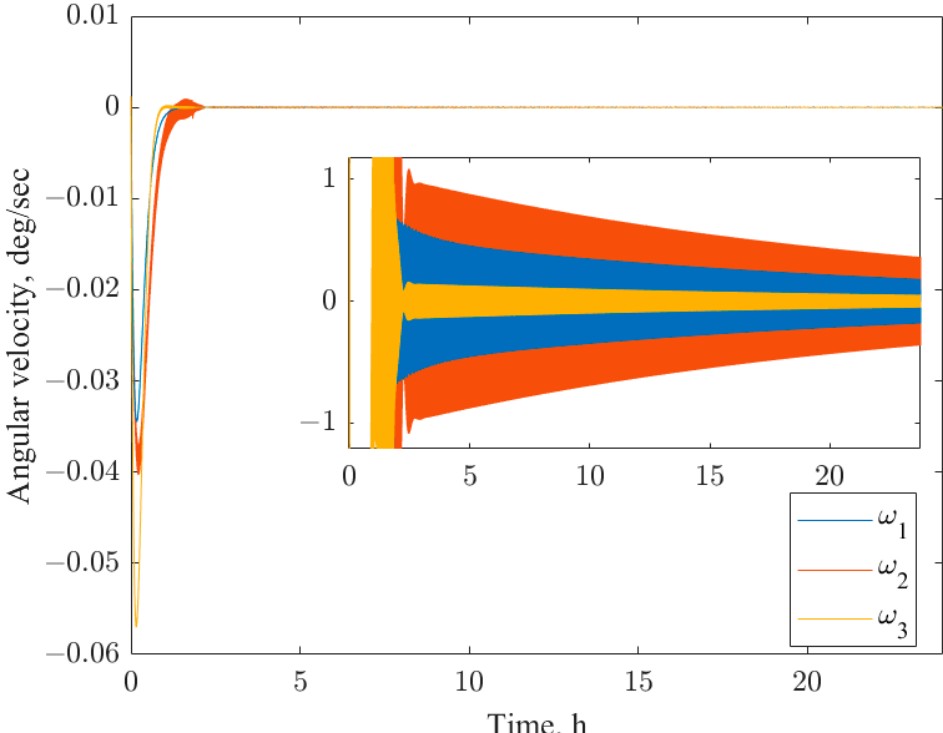

**Figure 4.** Angular velocity of the hub (*S*).

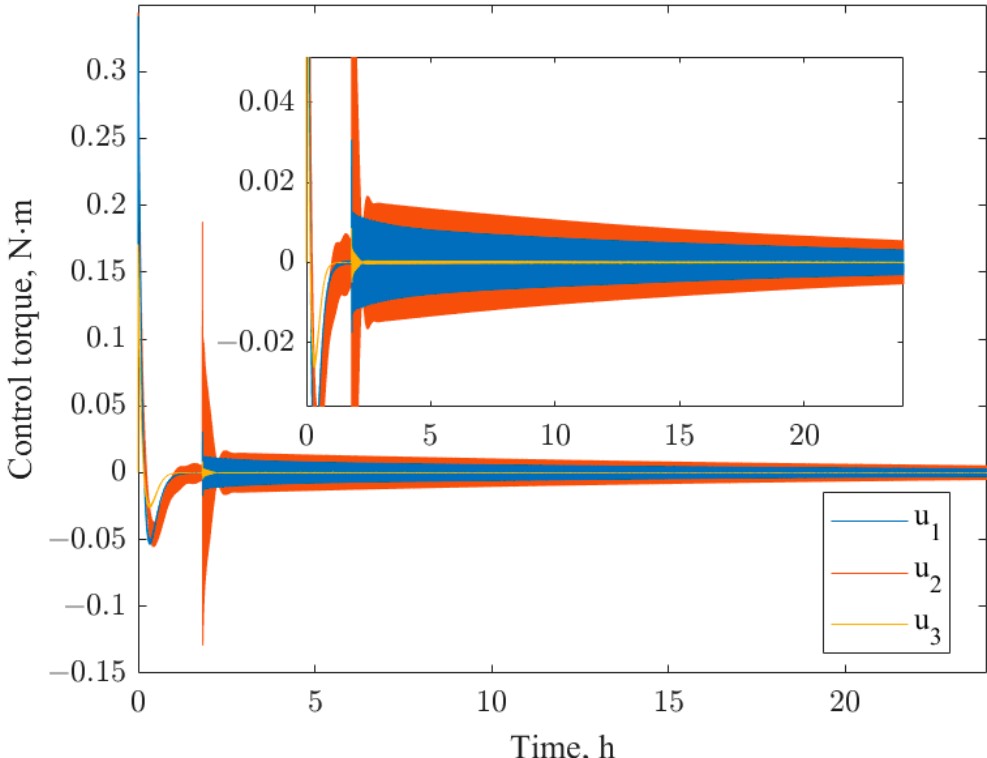

**Figure 5.** Reaction wheel control torque.

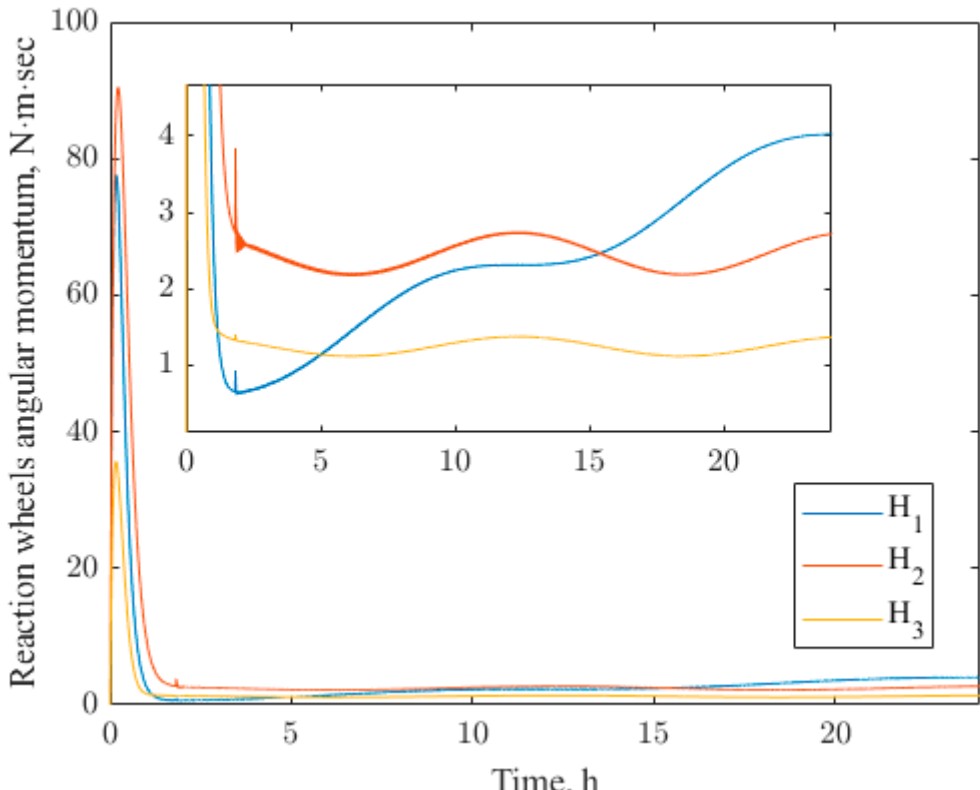

**Figure 6.** Reaction wheel angular momentum.

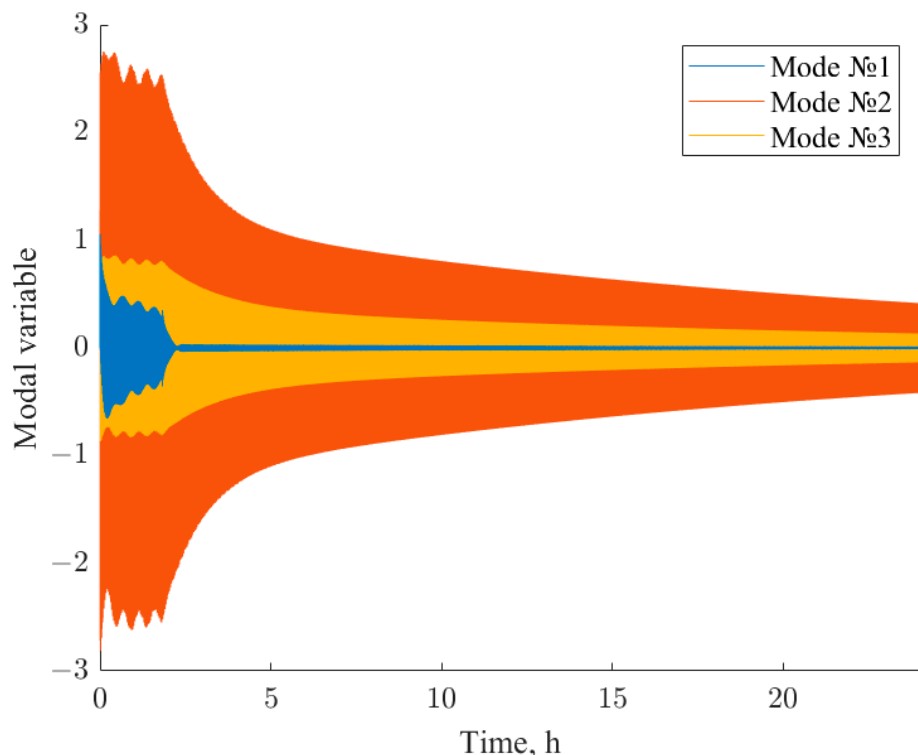

**Figure 7.** Modal variables of the antenna ($FE_3$).

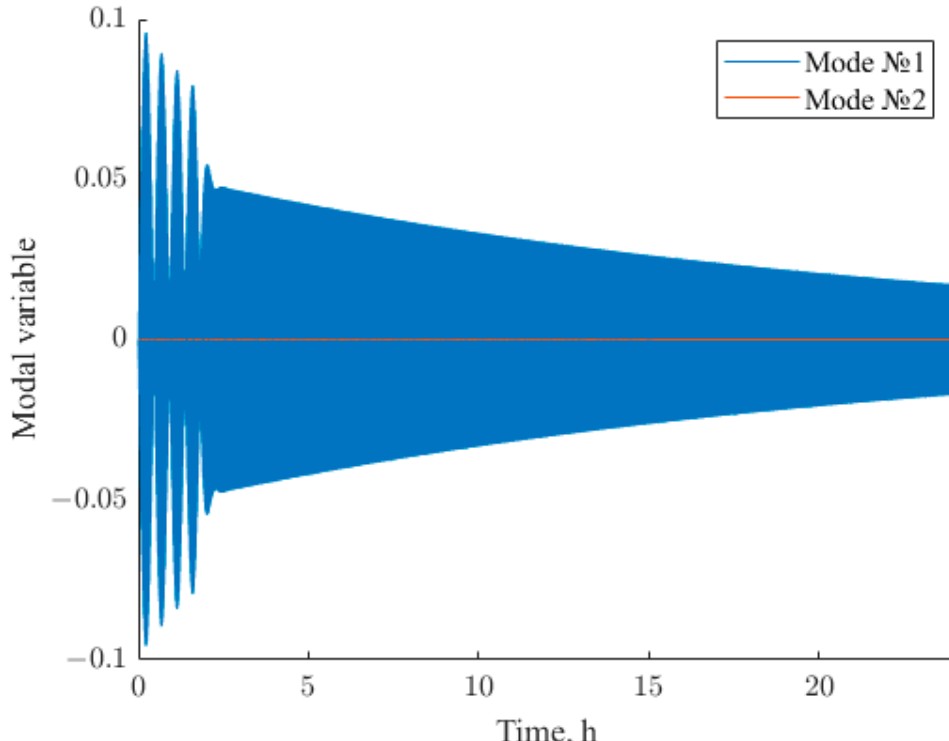

**Figure 8.** Model variables of one of the solar panels ($FE_1$).

**Table 2.** System parameters.

| $m_s$,kg | 3500 |
|---|---|
| $\mathbf{J}_s$, kg $\cdot$ m$^2$ | $diag(1.8 \quad 0.9 \quad 1.5) \times 10^3$ |
| $\mathbf{s}_1$, m | $(0 \quad 0 \quad -0.75)$ |
| $\mathbf{s}_2 = -\mathbf{s}_3$, m | $(0.5 \quad 0 \quad 0)$ |
| $m_1$, kg | 130 |
| $\mathbf{J}_1$, kg $\cdot$ m$^2$ | $\begin{pmatrix} 8 & 0 & 0 \\ 0 & 7 & -0.4 \\ 0 & -0.4 & 2 \end{pmatrix} \times 10^4$ |
| $\mathbf{f}_1$, m | $(0 \quad 2.5 \quad -12)$ |
| $m_{2,3}$, kg | 310 |
| $\mathbf{J}_{2,3}$, kg $\cdot$ m$^2$ | $\begin{pmatrix} 180 & -0.6 & 0 \\ -0.6 & 730 & 0 \\ 0 & 0 & 910 \end{pmatrix}$ |
| $\mathbf{f}_2 = -\mathbf{f}_3$, m | $(4 \quad 0 \quad 0)$ |

## 8. Conclusions

The stabilization of a satellite with large flexible elements by means of reaction wheels only is shown in the paper. The stabilizing control based on the LQR is provided. The sufficient condition for the asymptotic stability is derived. This condition has an explicit form and can be checked once the spacecraft configuration is known. The choice of control parameters is based on the closed form solution of Riccati's equation and the degree of stability of the system. The PSO usage allows one to find the parameters that give a rather good convergence rate and at the same time fulfill the control constraints.

**Author Contributions:** Study design and control synthesis, S.T.; mechanical model derivation, A.S.; particle swarm implementation and original draft preparation, A.O.; system parameter calculations, A.N. All authors have read and agreed to the published version of the manuscript.

**Funding:** This research received no external funding.

**Data Availability Statement:** Data are contained within the article.

**Conflicts of Interest:** The authors declare no conflict of interest.

## Appendix A

The d'Alembert principle for the whole system has the form (considering that constraints are ideal) [34]

$$\sum_{i \in S} \left( m_{s,i} \ddot{\mathbf{R}}_{s,i} - \mathbf{F}_{s,i} \right)^T \delta \mathbf{R}_{s,i} + \sum_{k=1}^{3} \left( \sum_{i \in FE_k} \left( m_{k,i} \ddot{\mathbf{R}}_{k,i} - \mathbf{F}_{k,i} \right)^T \delta \mathbf{R}_{k,i} \right) = 0. \tag{A1}$$

or

$$\left( \mathbf{S}_s \begin{pmatrix} \ddot{\mathbf{R}}_s \\ \dot{\boldsymbol{\omega}}_s \end{pmatrix} - \mathbf{N}_s \right)^T \begin{pmatrix} \delta \mathbf{R}_s \\ \delta \boldsymbol{\theta}_s \end{pmatrix} + \sum_{k=1}^{3} \left( \left( \mathbf{S}_k \begin{pmatrix} \ddot{\mathbf{R}}_k \\ \dot{\boldsymbol{\omega}}_k \\ \mathbf{q}_k \end{pmatrix} - \mathbf{N}_k \right)^T \begin{pmatrix} \delta \mathbf{R}_k \\ \delta \boldsymbol{\theta}_k \\ \delta \mathbf{q}_k \end{pmatrix} \right) = 0. \tag{A2}$$

Given that all three *FEs* are attached to the satellite hub in cantilever fashion, the angular velocities of the hub and each *FE* as well as their virtual rotations are congruent, i.e., $\forall k \, \delta \boldsymbol{\theta}_k = \delta \boldsymbol{\theta}_s$, $\boldsymbol{\omega}_k = \boldsymbol{\omega}_s$. The following independent virtual displacements are taken: the satellite hub center of mass position $\delta \mathbf{R}_s$, the rotation of the satellite $\delta \boldsymbol{\theta}_s$ and the flexible element modal coordinates $\delta \mathbf{q}_k$. By indicating with $\mathbf{s}_k$ the radius vector of the $FE_k$'s

mounting point relative to the hub's mass center and $\mathbf{f}_k$—the radius vector of the mass center of undeformed $FE_k$ relative to $FE_k$'s mounting point, we have

$$\begin{pmatrix} \delta\mathbf{R}_k \\ \delta\boldsymbol{\theta}_k \\ \delta\mathbf{q}_k \end{pmatrix} = \mathbf{W}_{k,1}\begin{pmatrix} \delta\mathbf{R}_s \\ \delta\boldsymbol{\theta}_s \end{pmatrix} + \mathbf{W}_{k,2}\delta\mathbf{q}_k, \tag{A3}$$

where

$$\mathbf{W}_{k,1} = \begin{pmatrix} \mathbf{E}_{3\times3} & -[\mathbf{s}_k + \mathbf{f}_k]_\times \\ \mathbf{0}_{3\times3} & \mathbf{E}_{3\times3} \\ \mathbf{0}_{n_k\times3} & \mathbf{0}_{n_k\times3} \end{pmatrix}, \quad \mathbf{W}_{k,2} = \begin{pmatrix} \mathbf{0}_{3\times n_k} \\ \mathbf{0}_{3\times n_k} \\ \mathbf{E}_{n_k\times n_k} \end{pmatrix}. \tag{A4}$$

Here, the expression $\delta\mathbf{R}_k = \delta\mathbf{R}_s + \delta\boldsymbol{\theta}_s \times (\mathbf{s}_k + \mathbf{f}_k)$ is used. The first and second derivatives of $\mathbf{R}_k$ are

$$\begin{aligned} \dot{\mathbf{R}}_k &= \dot{\mathbf{R}}_s + \boldsymbol{\omega}_s \times (\mathbf{s}_k + \mathbf{f}_k), \\ \ddot{\mathbf{R}}_k &= \ddot{\mathbf{R}}_s + \dot{\boldsymbol{\omega}}_s \times (\mathbf{s}_k + \mathbf{f}_k) + \boldsymbol{\omega}_s \times \boldsymbol{\omega}_s \times (\mathbf{s}_k + \mathbf{f}_k), \end{aligned} \tag{A5}$$

then, we obtain

$$\begin{pmatrix} \ddot{\mathbf{R}}_k \\ \dot{\boldsymbol{\omega}}_k \\ \ddot{\mathbf{q}}_k \end{pmatrix} = \mathbf{W}_{k,1}\begin{pmatrix} \ddot{\mathbf{R}}_s \\ \dot{\boldsymbol{\omega}}_s \end{pmatrix} + \mathbf{W}_{k,2}\ddot{\mathbf{q}}_k + \mathbf{T}_k, \tag{A6}$$

where $\mathbf{T}_k = \left( ((\boldsymbol{\omega}_s \times \boldsymbol{\omega}_s \times (\mathbf{s}_k + \mathbf{f}_k))^T \quad \mathbf{0}_{1\times3} \quad \mathbf{0}_{1\times n_k} \right)^T$. Finally, Equation (A2) takes the following form

$$\left(\mathbf{S}_s\begin{pmatrix} \ddot{\mathbf{R}}_s \\ \dot{\boldsymbol{\omega}}_s \end{pmatrix} - \mathbf{N}_s\right)^T\begin{pmatrix} \delta\mathbf{R}_s \\ \delta\boldsymbol{\theta}_s \end{pmatrix} + \sum_{k=1}^{3}\left(\mathbf{S}_k\left(\mathbf{W}_{k,1}\begin{pmatrix} \ddot{\mathbf{R}}_s \\ \dot{\boldsymbol{\omega}}_s \end{pmatrix} + \mathbf{W}_{k,2}\ddot{\mathbf{q}}_k + \mathbf{T}_k\right) - \mathbf{N}_k\right)^T\left(\mathbf{W}_{k,1}\begin{pmatrix} \delta\mathbf{R}_s \\ \delta\boldsymbol{\theta}_s \end{pmatrix} + \mathbf{W}_{k,2}\delta\mathbf{q}_k\right) = 0 \tag{A7}$$

The independence of generalized coordinates gives (10).

## Appendix B. General Force Calculation

*Appendix B.1. Gravity*

In the point mass model, the gravity force for the $i$-th point is [45]

$$\mathbf{F}_{k,i}^{grav} = -\frac{\mu m_{k,i}\mathbf{R}_{k,i}}{R_{k,i}^3}. \tag{A8}$$

Hereinafter, bold stands for the vector and regular for modulus of the vector $|\mathbf{R}_{k,i}| = R_{k,i}$. The hub general force vector is [46]

$$\begin{pmatrix} \mathbf{F}_S^{grav} \\ \mathbf{M}_S^{grav} \end{pmatrix} = \begin{pmatrix} -\mu m_S\dfrac{\mathbf{R}_S}{R_S^3} \\ \dfrac{3\mu}{R_S^5}\mathbf{R}_S \times \mathbf{J}_S\mathbf{R}_S \end{pmatrix}, \tag{A9}$$

where $\mu$ is the Earth gravity parameter. The corresponding vector for the flexible element is derived under $\boldsymbol{\rho}_{k,i} \ll \mathbf{r}_{k,i} \ll \mathbf{R}_{k,i}$. Thus,

$$\begin{aligned} \frac{\mathbf{R}_{k,i}}{R_{k,i}^3} &= \frac{1}{R_k^3}\frac{\mathbf{R}_k + (\mathbf{r}_{k,i} + \boldsymbol{\rho}_{k,i})}{\left(1 + 2\frac{\mathbf{R}_k(\mathbf{r}_{k,i} + \boldsymbol{\rho}_{k,i})}{R_k^2} + \frac{(\mathbf{r}_{k,i} + \boldsymbol{\rho}_{k,i})^2}{R_k^2}\right)^{3/2}} \approx \frac{1}{R_k^3}\frac{\mathbf{R}_k + (\mathbf{r}_{k,i} + \boldsymbol{\rho}_{k,i})}{\left(1 + \frac{\mathbf{R}_k(\mathbf{r}_{k,i} + \boldsymbol{\rho}_{k,i})}{R_k^2}\right)^3} \approx \\ &\approx \frac{1}{R_k^3}(\mathbf{R}_k + (\mathbf{r}_{k,i} + \boldsymbol{\rho}_{k,i}))\left(1 - 3\frac{(\mathbf{r}_{k,i} + \boldsymbol{\rho}_{k,i})^T\mathbf{R}_k}{R_k^2}\right). \end{aligned} \tag{A10}$$

Then,

$$
\begin{aligned}
\mathbf{F}_k^{grav} &= -\sum_i \mu m_{k,i} \frac{\mathbf{R}_{k,i}}{R_{k,i}^3} \approx \\
&\approx -\frac{\mu}{R_k^3}\sum_i m_{k,i}\big(\mathbf{R}_k + \mathbf{r}_{k,i} + \boldsymbol{\rho}_{k,i}\big)\left(1 - 3\frac{(\mathbf{r}_{k,i}+\boldsymbol{\rho}_{k,i})^T\mathbf{R}_k}{R_k^2}\right) \approx -\frac{\mu}{R_k^3}m_k\mathbf{R}_k,
\end{aligned}
\tag{A11}
$$

$$
\begin{aligned}
\sum_i \big(\mathbf{r}_{k,i}+\boldsymbol{\rho}_{k,i}\big)\times\mathbf{F}_{k,i}^{grav} &= -\mu\sum_i \big(\mathbf{r}_{k,i}+\boldsymbol{\rho}_{k,i}\big)\times m_{k,i}\frac{\mathbf{R}_k+(\mathbf{r}_{k,i}+\boldsymbol{\rho}_{k,i})}{\big|\mathbf{R}_k+(\mathbf{r}_{k,i}+\boldsymbol{\rho}_{k,i})\big|^3} = \\
&= -\mu\sum_i \frac{m_{k,i}(\mathbf{r}_{k,i}+\boldsymbol{\rho}_{k,i})\times\mathbf{R}_k}{\big|\mathbf{R}_k+(\mathbf{r}_{k,i}+\boldsymbol{\rho}_{k,i})\big|^3} \approx \\
&\approx -\frac{\mu}{R_k^3}\sum_i m_{k,i}(\mathbf{r}_{k,i}+\boldsymbol{\rho}_{k,i})\times\mathbf{R}_n\left(1-3\frac{(\mathbf{r}_{k,i}+\boldsymbol{\rho}_{k,i})^T\mathbf{R}_n}{R_n^2}\right) = \\
&= -\frac{\mu}{R_k^3}m_k\boldsymbol{\rho}_k\times\mathbf{R}_k + \frac{3\mu}{R_k^5}\sum_i m_i\big((\mathbf{r}_{k,i}+\boldsymbol{\rho}_{k,i})\times\mathbf{R}_k\big)(\mathbf{r}_{k,i}+\boldsymbol{\rho}_{k,i})^T\mathbf{R}_k = \\
&= -\frac{\mu}{R_k^3}m_k\boldsymbol{\rho}_k\times\mathbf{R}_k + \frac{3\mu}{R_k^5}\mathbf{R}_k\times\widetilde{\mathbf{J}}_k\mathbf{R}_k,
\end{aligned}
\tag{A12}
$$

$$
\begin{aligned}
\sum_i \mathbf{A}_{k,i}^T\mathbf{F}_{k,i}^{grav} &= -\mu\sum_i \mathbf{A}_{k,i}^T m_{k,i}\frac{\mathbf{R}_{k,i}}{R_{k,i}^3} \approx \\
&\approx -\frac{\mu}{R_k^3}\sum_i \mathbf{A}_{k,i}^T m_i\big(\mathbf{R}_k+\mathbf{r}_{k,i}+\boldsymbol{\rho}_{k,i}\big)\left(1-3\frac{(\mathbf{r}_{k,i}+\boldsymbol{\rho}_{k,i})^T\mathbf{R}_k}{R_k^2}\right) = \\
&= -\frac{\mu}{R_k^3}\sum_i \mathbf{A}_{k,i}^T m_{k,i}\big(\mathbf{R}_k+\mathbf{r}_{k,i}+\boldsymbol{\rho}_{k,i}\big) + 3\frac{\mu}{R_n^3}\sum_i \mathbf{A}_{k,i}^T m_{k,i}\mathbf{R}_k\frac{(\mathbf{r}_{k,i}+\boldsymbol{\rho}_{k,i})^T\mathbf{R}_k}{R_k^2} + \\
&\quad +3\frac{\mu}{R_k^3}\sum_i \mathbf{A}_{k,i}^T m_{k,i}\big(\mathbf{r}_{k,i}+\boldsymbol{\rho}_{k,i}\big)\frac{(\mathbf{r}_{k,i}+\boldsymbol{\rho}_{k,i})^T\mathbf{R}_k}{R_k^2}.
\end{aligned}
\tag{A13}
$$

Finally, under $\mathbf{u}_{n,i} \ll \mathbf{r}_{n,i} \ll \mathbf{R}_{n,i}$

$$
\boldsymbol{\Phi}_n^{grav} = \begin{pmatrix} -\frac{\mu}{R_k^3}m_k\mathbf{R}_k \\ -\frac{\mu}{R_k^3}m_k\boldsymbol{\rho}_k\times\mathbf{R}_k + \frac{3\mu}{R_n^5}\mathbf{R}_k\times\widetilde{\mathbf{J}}_k\mathbf{R}_k \\ -\frac{\mu}{R_k^3}m_k\mathbf{A}_k^T\mathbf{R}_k \end{pmatrix}.
\tag{A14}
$$

This value is used in the numerical simulation. Since modal variables are unknown when control is being synthesized, the rigid body part of this vector is used:

$$
\boldsymbol{\Phi}_n^{grav} = \begin{pmatrix} -\frac{\mu}{R_k^3}m_k\mathbf{R}_k \\ \frac{3\mu}{R_n^5}\mathbf{R}_k\times\mathbf{J}_k\mathbf{R}_k \\ \mathbf{0} \end{pmatrix}.
\tag{A15}
$$

It can be shown that under $\mathbf{u}_{n,i} \ll \mathbf{r}_{n,i} \ll \mathbf{R}_{n,i}$, the gravity gradient torque for the whole system becomes

$$
\mathbf{M}^{grav} = \frac{3\mu}{R^5}\mathbf{R}\times\mathbf{J}\mathbf{R},
\tag{A16}
$$

where $\mathbf{R}$ is the radius vector of the center of mass and $\mathbf{J}$ is the inertia tensor of the undeformed system.

*Appendix B.2. Solar Radiation Pressure*

Solar radiation pressure (SRP) is considered to affect the solar panels only, since the size of the hub is rather small, while the antenna has a lattice structure. Due to the small deformations, only the "rigid" part of the solar radiation force is taken into account. The force has the form [45]

$$
\mathbf{F}_k^{sun} = -S_k\frac{\Phi_0}{c}\big(\mathbf{n}_k^{sun}, \mathbf{n}_k^{pan}\big)\left((1-\alpha)\mathbf{n}_k^{sun} + 2\alpha\beta\big(\mathbf{n}_k^{sun}, \mathbf{n}_k^{pan}\big)\mathbf{n}_k^{pan} + \alpha(1-\beta)\Big(\mathbf{n}_k^{sun} + \frac{2}{3}\mathbf{n}_k^{pan}\Big)\right),
\tag{A17}
$$

where $S_k$ is the area of the panel, $\Phi_0 = 1367 \, W/m^2$ is the solar flux constant, $c$ is the speed of light, $\mathbf{n}_k^{sun}$ is the unit vector of the Sun direction (from satellite towards the Sun), $\mathbf{n}_k^{pan}$ is the panel normal, $\alpha$ and $\beta$ are the reflectivity and specularity coefficients. It is also considered here that $\left( \mathbf{n}_k^{sun}, \mathbf{n}_k^{pan} \right) \geq 0$. Each panel is considered to be symmetrical, so the net torque for each panel with respect to its center of mass is zero. Thus, the SRP effect is

$$\mathbf{\Phi}_n^{sun} = \begin{pmatrix} -S_k \frac{\Phi_0}{c} \left( \mathbf{n}_k^{sun}, \mathbf{n}_k^{pan} \right) \left( (1-\alpha)\mathbf{n}_k^{sun} + 2\alpha\beta \left( \mathbf{n}_k^{sun}, \mathbf{n}_k^{pan} \right) \mathbf{n}_k^{pan} + \alpha(1-\beta) \left( \mathbf{n}_k^{sun} + \frac{2}{3}\mathbf{n}_k^{pan} \right) \right) \\ 0 \\ 0 \end{pmatrix} \tag{A18}$$

Since the solar panels are almost identical and mounted, the symmetrical the net torque is almost zero.

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
