# Peer review of "Attitude Stabilization of a Satellite with Large Flexible Elements Using On-Board Actuators Only"

_mathematics, doi:10.3390/math11244928_

Round 1
Reviewer 1 Report
Comments and Suggestions for Authors
This paper develops a satellite's attitude control with three adjustable parts is examined. A set of reaction wheels fitted on the central hub of a satellite generates control torque. Because the flexible parts are considerable, the control torque constraints must be considered. The research investigates the control algorithm, which is based on the linear-quadratic regulator. This control's asymptotic stability is illustrated. The control parameters are chosen using the closed-form solution of the relevant algebraic Riccati problem, augmented by the linear matrix inequality. The particle swarm optimization is utilized to alter the control parameters to boost the convergence rate. General Comments:
1) In formula 15, why have you assumed that there is no torque and external force other than control, mention your reason
2) In all sections, the formulas you have used, please mention the reference. 3) Why have you examined only 3 *3 matrices in the Control synthesis section 5
Author Response
We would like to thank the Reviewer for his comments.
1. We have added comments in the begging of Section 5.
2. We have added the references for fundamental results on which our work is based.
3. We have added clarification on the page 8.
Reviewer 2 Report
Comments and Suggestions for Authors
1. To construct the attitude control and study the asymptotic stability of the desired equilibrium, the linearized equations of satellite motion relative to its center of mass are used in this paper. However, the linear modeling method is not the best approach to represent a nonlinear system. Some mathematic modeling methods, e.g., fuzzy modeling method, should be employed in this paper to represent the considered nonlinear system.
2. According to the linearized model, the linear-quadratic regulator (LQR) is used to stabilize the closed-loop system. However, the LQR approach is a well-known optimal control method that has been widely applied to many practical control systems. In this paper, there is no novel control approach can be found. The theoretical soundness of this paper is inferior.
3. Some elements of the matrices of the LQR parameters obtained in Equations (69) and (70) are huge. The significant elements in these parameters seem to induce infeasible control forces for the practical systems.
4. Considering the flow chart in Figure 2, some tuning directions for the parameters in many blocks are not provided. How do designers follow the direction to obtain the feasible solutions of the desired control gains? Besides, the convergence of the proposed design procedure was not discussed in this paper.
5. In summary, the control method employed in this paper is a traditional control approach. The control methodology was not developed or improved in this paper. In addition, the design process described in this paper is not straightforward. It is difficult for the readers to follow the design procedure to obtain feasible solutions for optimal control gains. The novelty and contribution of this paper are slight.
Comments on the Quality of English LanguageMinor editing of English language required.
Author Response
We are grateful to the Reviewer for his valuable comments.
- We agree that fuzzy modeling method may be used for the system modelling. However, here we use the classical approach based on the d'Alembert principle to obtain precise enough nonlinear model and by linearizing it we obtain the model which becomes the basis of the control law. In section 5 we prove that this approach allows to obtain the control which guarantee the asymptotic stability for equilibrium of the non-linear system.
- You are right that this control is well-known. However, in the present paper the control is based on the reduced model (for rigid body with fewer degrees of freedom) while being applied to the full one, which include deformation modes. Direct implementation of LQR in this case may lead to instability, so proper selection of the control parameters must be done. In section 5 the sufficient conditions of the asymptotic stability are derived and explicit solution of the algebraic Riccati equation (ARE) for this system is found. We have added this comment to the introduction.
- Yes, it is true. These matrices may lead to infeasible control efforts. We have calculated the feedback matrices and provide a remark on the page 18. Shortly, in section 6 we chose the control parameters such that for every initial conditions in the domains (67) and (68) the control will be less than 1 Nm which is feasible for the spacecraft like this. If the initial domain is more than the presented here, other sets of control parameters must be taken. The initial domain selection is described on the page 17.
- Thank you for your carefully study of the chart. We made some corrections to this chart. The remark of the convergence of the proposed design procedure is added to the page 14.
- In the introduction, we outlined the particularities of the LQR implementation in the paper. The methodology of overcoming the difficulties here we believe may be interesting of the journal readers.
Reviewer 3 Report
Comments and Suggestions for Authors
Attitude control of a satellite with three flexible elements is the hot topic in current research. The control algorithm which is based on the linear-quadratic regulator is studied. The asymptotic stability of this control is shown. The choice of the control parameters is based on the closed form solution of the corresponding algebraic Riccati equation, which is supplemented by the linear matrix inequality. To increase the convergence rate, the particle swarm optimization is used to tune the control parameters. As for me, there are still many areas that need improvement, as follows:
1. Experimental results should be added for verification to further confirm the correctness and feasibility of the paper.
2. There are grammar errors in some parts of the text, please check them, such as Line 255.
Author Response
We would like to thank the Reviewer for his comments.
1. In the paper, we have focused on the mathematical aspects of the problem. By now the correctness of the approach is based on the correct application of the control and stability theory. The numerical example shows the specific system behavior. The control feasibility is clarified on the pages 17 and 18.
2. We corrected this typo and also reread the whole paper.
Reviewer 4 Report
Comments and Suggestions for Authors
My comments are attached to the attachment. Please check.

The English language should be polished again.
Author Response
We are grateful to the Reviewer. Your remarks allow us to improve our paper.
1. We add several paragraphs to the Introduction (see red text).
2-4. We have corrected typos.
5. We have added clarification in this section. See page 24.
Round 2
Reviewer 2 Report
Comments and Suggestions for Authors
1. The authors have made a few modifications to this paper. Just some tiny additions were made. There is no increase in the main substantive content of the paper.
2. The authors did not respond positively to my comments and did not make corresponding content improvements. I can’t see any improvement in the paper after the revision.
3. The authors agree with my comment that the matrices of the conditions may lead to infeasible control efforts. However, the authors did not provide any revisions for the conditions developed in this paper. The contribution of this paper cannot be verified.
Comments on the Quality of English Language
Minor editing of English language required.
Author Response
We would like to thank the Reviewer for his comments.
- We added clarifications to the Introduction and Sections of the paper, and outlined the crucial topics of our research. We also corrected the typos in text, formulas and figures. Concerning the general remark about using d'Alembert principle and classical mechanics approach together with LQR instead of the fuzzy logic modelling, that would be a different article in this case. Comparing these approaches could be the topic of another paper.
- Unfortunately, we cannot completely agree with you here. We responded to every comment and four out of five comments received, as we suppose, positive feedback. The exception was the first comment to demand a change in the content of the paper. The problem was not only to solve the stabilization problem, but also to figure out why and when the well-known and easy-to-implement algorithm could solve the problem. While the LQR is already a classical algorithm, the situation of its application in the paper is non-trivial. The clarifications were added to the Introduction. We agreed that a fuzzy logic approach, kindly suggested by the Reviewer, could solve the problem, but this is a different approach, and it is beyond the scope of our paper.
- Yes, we agreed and additionally clarified this in the response to Reviewer and in the paper on the pages 17 and 18. The presented linear feedback matrices can only lead to the infeasible control when the initial conditions are too far from the desired equilibrium, which was further specified in the paper on the page 18. In Section 6, it was proven that both of the presented feedback matrices will not lead to infeasible control if the initial state is in the mentioned domain. The system will not leave this area if the presented matrices are used in control. We now have added a little explanation on the page 14.
Reviewer 3 Report
Comments and Suggestions for Authors
The paper is now improved. I am satisfied with the changes
Author Response
Thank you for your positive feedback.